# LMA: Latent Motion Adjuster for Physics-based Multi-agent Interaction

**Taiki Yano**  *yano.taiki.34u@st.kyoto-u.ac.jp*
*Kyoto University*

**Satoshi Yagi**  *yagi@i.kyoto-u.ac.jp*
*Kyoto University*

**Satoshi Yamamori**  *yamamori@lm.sys.i.kyoto-u.ac.jp*
*Kyoto University*

**Jun Morimoto**  *morimoto@i.kyoto-u.ac.jp*
*Kyoto University*

**Reviewed on OpenReview:** *https://openreview.net/forum?id=OR5ClOTaWU*

## Abstract

Learning interactive multi-agent behaviors from scratch is often sample-inefficient and fails to exploit reusable skills learned in simpler settings. While latent skill representations enable efficient single-agent reinforcement learning, their extension to multi-agent interaction requires conditioning behaviors on other agents without destroying pretrained structure. We formulate multi-agent interaction as a latent adaptation problem and propose the Latent Motion Adjuster (LMA), a lightweight conditional module that modifies latent actions produced by a pretrained single-agent policy based on other agents' states. Rather than relearning policies from scratch, our method performs structured residual adaptation in latent space, enabling efficient skill reuse under both cooperative and competitive scenarios. Experiments on physics-based control benchmarks demonstrate that latent-space adaptation improves sample efficiency and interaction performance over fine-tuning and strategic baselines. These results suggest that conditional latent modulation provides a principled mechanism for transferring single-agent skills to multi-agent reinforcement learning.

## 1 Introduction

Learning physically consistent control policies for embodied agents remains a central challenge in reinforcement learning. Early approaches to character and robot control relied on carefully designed dynamic controllers (Ijspeert, 2008; Raibert & Hodgins, 1991), imitation learning from human demonstrations (Schmidts et al., 2011; Schaal, 1999), or end-to-end reinforcement learning (Levine et al., 2016; Johannink et al., 2019). Although effective in specific tasks, these methods typically require task-specific training and often struggle to maintain natural, physically plausible behaviors under distributional shift.

Recent advances in hierarchical reinforcement learning address this limitation by introducing motion priors, i.e., pretrained latent skill representations that capture diverse and physically valid behaviors. Such representations have demonstrated strong transferability across downstream tasks in character animation (Peng et al., 2022; Tessler et al., 2023), VR avatar control (Luo et al., 2023; 2024a), and real robot control (Serifi et al., 2024). By learning compact latent spaces aligned with physical constraints, high-level policies can efficiently adapt to new objectives while preserving motion realism. Extensions of these ideas to multi-agent settings have also been explored; however, most prior approaches learn interaction policies from scratch in the multi-agent phase (Zhu et al., 2023; Liu et al., 2022; Won et al., 2022), which limits sample efficiency and reduces the benefit of previously acquired single-agent skills.

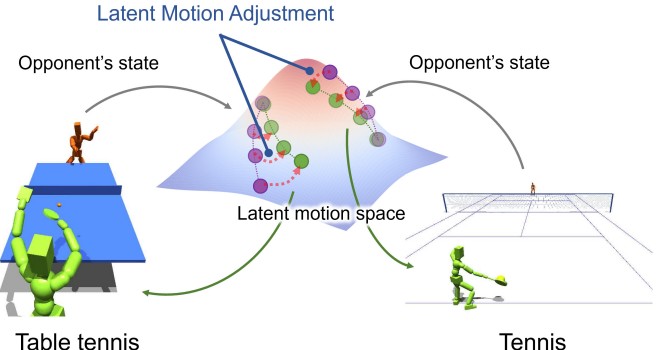

Figure 1: Overview of Latent Motion Adjuster (LMA): By adjusting motions represented in the latent space based on the observed behaviors of other agents, this approach leverages skills learned in the single-agent setting while enabling advanced cooperation and competition in multi-agent interactions.

In this work, we reformulate multi-agent interaction as a latent adaptation problem. Rather than relearning high-level policies for interactive settings, we propose to reuse single-agent skills and condition them on other agents through structured modulation in latent space. Concretely, we introduce the Latent Motion Adjuster (LMA), a lightweight conditional module that refines latent actions produced by a pretrained single-agent policy based on the observed state of other agents. This design enables efficient skill reuse while allowing policies to dynamically respond to cooperative or adversarial behavior.

We instantiate the proposed framework on physics-based control benchmarks involving whole-body humanoid agents in cooperative and competitive scenarios. Empirical results demonstrate that latent-space adaptation achieves stronger interaction performance and improved learning efficiency compared to fine-tuning or learning-from-scratch baselines. These findings suggest that conditional latent modulation provides a principled mechanism for transferring single-agent skills to multi-agent reinforcement learning.

The main contributions of this work are summarized as follows:

- We formulate multi-agent interaction as a latent skill adaptation problem and propose a multi-agent learning framework that reuses pretrained single-agent policies instead of learning interaction policies from scratch.

- We introduce the Latent Motion Adjuster (LMA), a lightweight conditional module that performs structured residual modulation in latent space, enabling policies to adapt to other agents while preserving pretrained skill structure.

- We empirically demonstrate that conditional latent adaptation improves interaction performance and sample efficiency compared to fine-tuning and learning-from-scratch baselines in physics-based multi-agent control benchmarks.

## 2 Related Work

### 2.1 Motion Prior

With the advancement of motion capture technology and human motion estimation from videos, numerous large-scale datasets covering diverse human motions have been proposed (Mahmood et al., 2019; Sigal et al., 2010). The motion prior is a motion generator that learns compact latent motion representations capable of reproducing such diverse human motions in a simulation. The pretrained motion prior is subsequently reused as a controller to generate motions for downstream tasks. By learning high-level policies to control

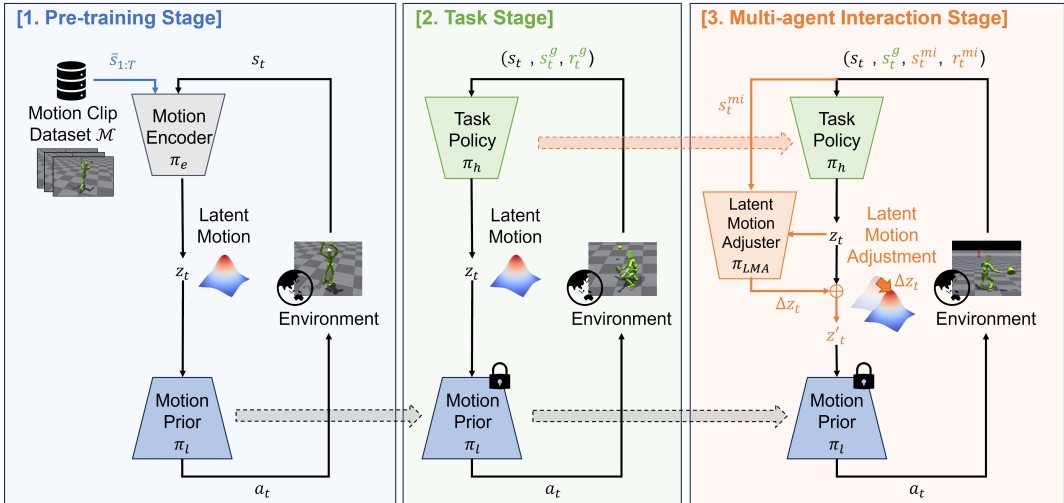

Figure 2: Structure and training flow of the proposed method. 1: Pre-training stage where the latent motion representation of the motion prior is acquired by imitating the motion clip dataset. 2: Task stage where the high-level policy is learned to control the latent motion of the motion prior according to the given task. 3: Multi-agent interaction stage where the latent motion adjuster is trained to adjust the latent motion output by the high-level policy according to the opponent's state.

the latent motions, it can quickly acquire effective actions for target tasks such as striking a target (Luo et al., 2024a; Peng et al., 2022; Tessler et al., 2023), javelin throwing (Luo et al., 2024b), or waltz dancing (Cui et al., 2024).

Several prior studies have proposed the use of the motion prior not only for single-agent control but also for multi-agent interactions (Zhu et al., 2023; Liu et al., 2022; Won et al., 2022; Han et al., 2024). However, these studies typically learn multi-agent interactions from scratch and do not leverage high-level policies acquired by a single agent, as our research does. The study by Han et al. (2024) is similar to our method in that it reuses single-agent policies for multi-agent interactions. However, it learns a strategic-level policy that outputs sub-goals for single-agent policies, and relying solely on adjusting such abstract sub-goals may hinder flexible adaptation to the opponent's state. In contrast, our method directly adjusts the latent motions generated by the single-agent policies, enabling more fine-grained behavioral modifications.

## 2.2 Residual Reinforcement Learning

Residual reinforcement learning (residual RL) enables data-efficient policy improvement by augmenting an existing controller with a learned corrective policy. Prior work has demonstrated the effectiveness of learning such corrective policies for a wide range of baseline controllers, including model predictive control–based controllers (Johannink et al., 2019), reinforcement-learning policies (Silver et al., 2019), and controllers obtained through imitation learning from expert trajectories (Rana et al., 2022; Ankile et al., 2025). However, because these approaches compute residuals directly in the action space, combining the learned corrections with the underlying controller can introduce instability. In contrast, our method computes residuals within a latent space that captures a diverse repertoire of behaviors, thereby reducing the instability associated with residual addition. Moreover, by learning residuals in this compact latent representation, the resulting corrective policy can be immediately applied to new tasks, enabling rapid and robust adaptation.

## 2.3 Multi-agent Reinforcement Learning Utilizing Skill Transfer

The acquisition of cooperative and adversarial policies among multiple agents has been a long-standing challenge within the framework of multi-agent reinforcement learning (MARL) (Zhang et al., 2021; Yu et al., 2022). Generally, MARL involves learning policies for multiple agents simultaneously. On the other hand,

many methods have also been proposed to leverage past experiences for immediate responses to multi-agent interactions. Won et al. (2021) developed a mixture-of-experts learning framework that adapts to tasks requiring multi-agent interactions by combining skills acquired through prior imitation learning. Haarnoja et al. (2024) achieved dynamic skill switching in soccer using actual humanoid robots through a two-stage learning process involving single-agent skill acquisition and multi-agent interactions via self-play. However, these methods require the preparation of multiple skills in advance, making it challenging to respond to multi-agent interactions by adjusting only a single skill, such as a motion prior.

## 3 Methods

The training process of the proposed method consists of three stages: 1) Pre-training stage: A motion prior is trained to acquire a latent motion representation by imitating a diverse set of motion clips. 2) Task stage: A high-level policy is learned to control the latent motion of the motion prior in order to solve specific downstream tasks. 3) Multi-agent interaction stage: A latent motion adjuster is trained to modulate the latent motions generated by the high-level policy based on the state of an opponent (Fig. 2).

The overall policies are trained using a VAE-based model architecture (Kingma & Welling, 2014), with each stage contributing to the progressive construction of the final controller. Details of each stage are described in the following sections.

### 3.1 Learning Latent Motion Representation of Motion Prior

In the pre-training stage, the motion encoder and the motion prior are trained to imitate various character's motions (left of Fig. 2). The motion encoder $\pi_e(\mathbf{z}_t|\mathbf{s}_t, \bar{\mathbf{s}}_{1:T})$ generates latent motions $\mathbf{z}_t \in \mathbb{R}^{d_z}$ from the agent's proprioceptive state $\mathbf{s}_t \in \mathbb{R}^{d_s}$ and the reference motion clip $\bar{\mathbf{s}}_{1:T}$ sampled from the motion clip dataset $\mathcal{M}$. Then, the motion prior $\pi_l(\mathbf{a}_t|\mathbf{s}_t, \mathbf{z}_t)$ outputs actions $\mathbf{a}_t \in \mathbb{R}^{d_a}$ from the agent's state $\mathbf{s}_t$ and the latent variable $\mathbf{z}_t$. Here, $\mathbf{s}_t$ and $\bar{\mathbf{s}}_{1:T}$ typically represent ego-centric states that include the positions and velocities of the character's body parts. The environment transitions to the next state $\mathbf{s}_{t+1}$ based on the transition probability $p(\mathbf{s}_{t+1}|\mathbf{s}_t, \mathbf{a}_t)$.

This study specifically follows the PULSE training flow (Luo et al., 2024a). In PULSE, a controller called PHC+ is separately prepared and trained to imitate the motions in the motion clip dataset. PHC+ achieves high performance, but directly outputs actions $\mathbf{a}_t^{\mathrm{PHC+}}$ from $\pi_{\mathrm{PHC+}}(\mathbf{a}_t^{\mathrm{PHC+}}|\mathbf{s}_t, \bar{\mathbf{s}}_{1:T})$ without utilizing latent motion representations. To address this, PULSE treats the actions output by PHC+ as teacher data and trains a motion encoder and decoder to mimic the actions of PHC+ minimizing the following reconstruction error $\mathcal{L}$,

$$\begin{aligned} \mathcal{L} = &||\mathbf{a}_t^{\mathrm{PHC+}} - \mathbf{a}_t||^2 + \alpha\mathcal{L}_{reg} + \\ &\beta D_{KL}[\pi_e(\mathbf{z}_t|\mathbf{s}_t, \bar{\mathbf{s}}_{1:T})||\mathcal{R}(\mathbf{z}_t|\mathbf{s}_t)], \end{aligned} \tag{1}$$

where $\mathcal{L}_{reg}$ is an additional regularization term for consecutive latent motions, $\mathcal{R}(\mathbf{z}_t|\mathbf{s}_t)$ is the prior distributions of the latent motions, and $\alpha$ and $\beta$ are hyperparameters. When the encoder distribution is modeled as a normal distribution $\mathcal{N}(\mathbf{z}_t|\boldsymbol{\mu}_t, \boldsymbol{\sigma}_t)$, $\mathcal{L}_{reg}$ constrains the distance between the centers $||\boldsymbol{\mu}_t - \boldsymbol{\mu}_{t-1}||^2$. PULSE also uses a learned conditional prior distribution $\mathcal{R}(\mathbf{z}_t|\mathbf{s}_t)$ in the KL constraint instead of a zero-mean Gaussian prior. By pretraining the motion prior through these interactions, the policy capable of generating diverse actions included in the motion clip dataset $\mathcal{M}$, and a latent space that represents these motions can be acquired. Algorithm 1 outlines the pre-training stage.

### 3.2 Learning Latent Motion Control for Downstream Task

In the task stage, we learn a high-level policy $\pi_h(\mathbf{z}_t|\mathbf{s}_t, \mathbf{s}_t^g)$ that outputs the latent variable $\mathbf{z}_t$ of the motion prior on the basis of the agent's state $\mathbf{s}_t$ and task-dependent state $\mathbf{s}_t^g$ (middle of Fig. 2). The motion prior is fixed, and the high-level policy is trained to maximize the following cumulative rewards,

$$\mathcal{J}_g = \mathbb{E}_{p(\boldsymbol{\tau})}^{\pi_h}\left[\sum_{t=1}^T \gamma^{t-1} r_t^g\right], \tag{2}$$

---

**Algorithm 1** Pre-training stage

---

**Input**: Motion clip dataset $\mathcal{M}$, pretrained PHC+ $\pi_{PHC+}$, encoder $\pi_e$, motion prior $\pi_l$, and prior distribution $\mathcal{R}$

1: **while** not converged **do**
2:     $\mathcal{D} \leftarrow \emptyset$
3:     **while** $\mathcal{D}$ not full **do**
4:         $\bar{\mathbf{s}}_{1:T} \sim \mathcal{M}$
5:         **for** $t \leftarrow 1 \ldots T$ **do**
6:             $\mathbf{z}_t \sim \pi_e(\mathbf{z}_t | \mathbf{s}_t, \bar{\mathbf{s}}_{1:T})$
7:             $\mathbf{a}_t \sim \pi_l(\mathbf{a}_t | \mathbf{s}_t, \mathbf{z}_t)$
8:             $\mathbf{a}_t^{\text{PHC+}} \sim \pi_{\text{PHC+}}(\mathbf{a}_t^{\text{PHC+}} | \mathbf{s}_t, \bar{\mathbf{s}}_{1:T})$
9:             store $(\mathbf{s}_t, \mathbf{z}_t, \mathbf{a}_t, \mathbf{a}_t^{\text{PHC+}})$ into $\mathcal{D}$
10:            $\mathbf{s}_{t+1} \sim p(\mathbf{s}_{t+1} | \mathbf{s}_t, \mathbf{a}_t)$
11:         **end for**
12:     **end while**
13:     Update $\pi_e, \pi_l, \mathcal{R}$ using pairs of $(\mathbf{s}_t, \mathbf{z}_t, \mathbf{a}_t, \mathbf{a}_t^{\text{PHC+}})$ and Equation 1
14: **end while**
15: **return** $\pi_e, \pi_l, \mathcal{R}$

---

**Algorithm 2** Task stage

---

**Input**: pretrained motion prior $\pi_l$, and high-level policy $\pi_h$

1: **while** not converged **do**
2:     $\mathcal{D} \leftarrow \emptyset$
3:     **while** $\mathcal{D}$ not full **do**
4:         **for** $t \leftarrow 1 \ldots T$ **do**
5:             $\mathbf{z}_t \sim \pi_h(\mathbf{z}_t | \mathbf{s}_t, \mathbf{s}_t^g)$
6:             $\mathbf{a}_t \sim \pi_l(\mathbf{a}_t | \mathbf{s}_t, \mathbf{z}_t)$
7:             $\mathbf{s}_{t+1}, \mathbf{s}_{t+1}^g \sim p(\mathbf{s}_{t+1}, \mathbf{s}_{t+1}^g | \mathbf{s}_t, \mathbf{s}_t^g, \mathbf{a}_t)$
8:             $r_t^g \leftarrow r_g(\mathbf{s}_t, \mathbf{s}_t^g, \mathbf{a}_t)$
9:             store $(\mathbf{s}_t, \mathbf{s}_t^g, \mathbf{z}_t, \mathbf{a}_t, r_t^g, \mathbf{s}_{t+1}, \mathbf{s}_{t+1}^g)$ into $\mathcal{D}$
10:         **end for**
11:     **end while**
12:     Update $\pi_h$ using Equation 2 and experiences collected in $\mathcal{D}$
13: **end while**
14: **return** $\pi_h$

---

where $r_t^g$ is the task-dependent reward $r_t^g = r_g(\mathbf{s}_t, \mathbf{s}_t^g, \mathbf{a}_t)$, $\mathbb{E}_{p(\boldsymbol{\tau})}^{\pi_h}[\cdot]$ is the expected value over the trajectory $\boldsymbol{\tau} = [\mathbf{s}_1, \mathbf{s}_1^g, \mathbf{a}_1, r_1^g, \cdots, \mathbf{s}_{T-1}, \mathbf{s}_{T-1}^g, \mathbf{a}_{T-1}, r_{T-1}^g, \mathbf{s}_T, \mathbf{s}_T^g]$ by following $\pi_h$, and $\gamma$ is the discount factor. By training the high-level policy in the task stage, it becomes possible to leverage the natural motion generation of the motion prior while acquiring actions suitable for the target task. Algorithm 2 outlines the task stage.

### 3.3 Latent Motion Adjustment for Multi-agent Interaction

Similar to the task stages discussed in the previous section, the goal of the agents in the multi-agent interaction stage is to maximize the following cumulative rewards,

$$\mathcal{J}_{mi} = \mathbb{E}_{p(\boldsymbol{\tau})}^{\pi_h} \left[ \sum_{t=1}^{T} \gamma^{t-1} r_t^{mi} \right], \tag{3}$$

where $r_t^{mi} = r_{mi}(\mathbf{s}_t, \mathbf{s}_t^g, \mathbf{s}_t^{mi}, \mathbf{a}_t)$ is the reward that reflect the quality of interactions among multiple agents. However, unlike when learning single-agent tasks, agents are required to output appropriate actions $\mathbf{a}_t$ based on the observed opponent's states $\mathbf{s}_t^{mi}$.

---

**Algorithm 3** Multi-agent interaction stage

---

**Input**: pretrained motion prior $\pi_l$ and high-level policy $\pi_h$, latent motion adjuster $\pi_{LMA}$, hyperparameter $\eta$, and freeze-high-level-policy

1: **while** not converged **do**
2:     $\mathcal{D} \leftarrow \emptyset$
3:     **while** $\mathcal{D}$ not full **do**
4:         **for** $t \leftarrow 1 \ldots T$ **do**
5:             $\mathbf{z}_t \sim \pi_h(\mathbf{z}_t | \mathbf{s}_t, \mathbf{s}_t^g)$
6:             $\Delta\mathbf{z}_t \sim \pi_{LMA}(\Delta\mathbf{z}_t | \mathbf{s}_t^{mi}, \mathbf{z}_t)$
7:             $\mathbf{z}'_t \leftarrow \mathbf{z}_t + \eta\Delta\mathbf{z}_t$
8:             $\mathbf{a}_t \sim \pi_l(\mathbf{a}_t | \mathbf{s}_t, \mathbf{z}'_t)$
9:             $\mathbf{s}_{t+1}, \mathbf{s}_{t+1}^g, \mathbf{s}_{t+1}^{mi} \sim p(\mathbf{s}_{t+1}, \mathbf{s}_{t+1}^g, \mathbf{s}_{t+1}^{mi} | \mathbf{s}_t, \mathbf{s}_t^g, \mathbf{s}_t^{mi}, \mathbf{a}_t)$
10:            $r_t^{mi} \leftarrow r_{mi}(\mathbf{s}_t, \mathbf{s}_t^g, \mathbf{s}_t^{mi}, \mathbf{a}_t)$
11:            store $(\mathbf{s}_t, \mathbf{s}_t^g, \mathbf{s}_t^{mi}, \mathbf{z}_t, \Delta\mathbf{z}_t, \mathbf{a}_t, r_t^{mi}, \mathbf{s}_{t+1}, \mathbf{s}_{t+1}^g, \mathbf{s}_{t+1}^{mi})$ into $\mathcal{D}$
12:         **end for**
13:     **end while**
14:     Update $\pi_{LMA}$ using Equation 3 and experiences collected in $\mathcal{D}$
15:     **if not** freeze-high-level-policy **then**
16:         Update $\pi_h$ using Equation 3 and experiences collected in $\mathcal{D}$
17:     **end if**
18: **end while**
19: **return** $\pi_{LMA}$, $\pi_h$

---

While prior researches generally involve learning high-level policies from scratch for multi-agent interactions, the proposed method aims to efficiently acquire policies for multi-agent interactions by leveraging skills obtained from single-agent tasks (Right of Fig. 2). The challenge of reusing single-agent skills in this manner lies in addressing the differences in observations received between single-agent tasks and multi-agent interaction tasks. As previously mentioned, in multi-agent interactions, it is necessary to consider the opponent's states $\mathbf{s}_t^{mi}$ in addition to the states received from the environment in single-agent tasks $(\mathbf{s}_t, \mathbf{s}_t^g)$. High-level policies acquired from single-agent tasks cannot directly input $\mathbf{s}_t^{mi}$ as they are. However, deciding actions without using $\mathbf{s}_t^{mi}$ makes it difficult to appropriately adjust actions based on the current states of the opponent, thereby hindering advanced cooperation and adversarial motions.

To address the challenge of reusing single-agent skills, we add a small model called a latent motion adjuster $\pi_{LMA}$ that can fine-tune the output of the high-level policy based on the opponent's states (Right side of Fig. 2). As shown in the following equation, $\pi_{LMA}$ outputs an increment $\Delta\mathbf{z}_t$ to adjust the latent motion based on the latent motion $\mathbf{z}_t$ output by the high-level policy $\pi_h$ and the current state of the opponent $\mathbf{s}_t^{mi}$,

$$\Delta\mathbf{z}_t \sim \pi_{LMA}(\Delta\mathbf{z}_t | \mathbf{s}_t^{mi}, \mathbf{z}_t). \tag{4}$$

The latent motion is adjusted to $\mathbf{z}'_t = \mathbf{z}_t + \eta\Delta\mathbf{z}_t$, where $\eta$ is a hyperparameter that determines the degree of adjustment for latent motion. Then, by inputting the adjusted latent motion $\mathbf{z}'_t$ into the motion prior $\pi_l$, the agent generates action $\mathbf{a}_t$ that consider the opponent's state. By learning the adjustment method of $\pi_{LMA}$ according to Equation 4, the proposed method can effectively utilize skills acquired from single-agent tasks while generating advanced cooperative and adversarial motions for multi-agent interactions. In the multi-agent interaction stage, the weights of the high-level policy may either be fixed or further updated. The former enables rapid adaptation, whereas the latter allows the acquisition of more sophisticated behaviors. Algorithm 3 outlines the multi-agent interaction stage.

## 4 Experiments

### 4.1 Environments

We used simulated table tennis and tennis environments in SMPLOlympics (Luo et al., 2024b), in which a physics simulator IsaacGym (Makoviychuk et al., 2021) is used to control agents (Fig. 1). The proprioceptive state $\mathbf{s}_t = [h_{root}, \mathbf{p}_{body}, \mathbf{v}_{body}, \boldsymbol{\theta}_{body}, \boldsymbol{\omega}_{body}] \in \mathbb{R}^{358}$ consisted of the height of the agent's pelvis $h_{root} \in \mathbb{R}^1$ and the position $\mathbf{p}_{body} \in \mathbb{R}^{3 \times (24-1)}$, velocity $\mathbf{v}_{body} \in \mathbb{R}^{3 \times 24}$, 6DoF rotation representation $\boldsymbol{\theta}_{body} \in \mathbb{R}^{6 \times 24}$, and rotational velocity $\boldsymbol{\omega}_{body} \in \mathbb{R}^{3 \times 24}$ of 24 body parts as seen from the pelvis. The rotation representation uses Zhou et al. (2019). That is, we convert rotation matrices into a continuous 6D representation through an orthogonalization process. Action $\mathbf{a}_t \in \mathbb{R}^{3 \times 23}$ is target joint angles for 23 joints, which are converted into torques using PD control.

In the task stage, $\mathbf{s}_t^g = [\mathbf{p}_{ball}, \mathbf{v}_{ball}, \mathbf{p}_{racket}, \mathbf{p}_{self}^*, \mathbf{p}_{ball}^*] \in \mathbb{R}^{15}$ consisted of the ball position $\mathbf{p}_{ball} \in \mathbb{R}^3$, ball velocity $\mathbf{v}_{ball} \in \mathbb{R}^3$, racket position $\mathbf{p}_{racket} \in \mathbb{R}^3$, the agent's target positions at the time of ball impact $\mathbf{p}_{self}^* \in \mathbb{R}^3$, and the target position for the ball's landing $\mathbf{p}_{ball}^* \in \mathbb{R}^3$. We used the reward $r_t^g$ designed in SMPLOlympics, which is intended to encourage hitting the ball back to the target position on the opponent's court.

In the multi-agent interaction stage, we conducted experiments with two types of tasks: rallies for cooperation and competitive matches. In both tasks, $\mathbf{s}_t^{mi} = [\mathbf{p}_{op}, \mathbf{p}_{racket\_op}] \in \mathbb{R}^6$ consisted of the positions of the opponent $\mathbf{p}_{op} \in \mathbb{R}^3$ and opponent's racket $\mathbf{p}_{racket\_op} \in \mathbb{R}^3$ as seen from the agent. In the rally task, we used $r_t^g$ as the reward $r_t^{mi}$. The cumulative reward of $r_t^g$ increases the longer the rally with the opponent continues, thus encouraging the acquisition of cooperative motions that return the ball in a way that is easy for the opponent to hit. On the other hand, in the match task, the reward $r_t^{mi}$ was a combination of $r_t^g$ and the match reward $r_{match}$, which is designed to encourage the acquisition of adversarial motions to defeat the opponent by giving a significantly larger reward compared to $r_t^g$ when winning the match and a penalty when losing the match. In this experiment, we assigned a reward $r_{match}$ of 100 for a win and a penalty of $-100$ for a loss. Further details on the environment and reward design are provided in Appendix A.1 and Appendix A.2, respectively.

While the observations and rewards for table tennis and tennis are similar, the skills required for learning differ significantly. In table tennis, minimal lower-body movement is required, and the primary skill involves returning the ball using only arm control. In contrast, tennis requires moving to the ball's landing position and returning it using the entire body. By evaluating the proposed method across these qualitatively distinct tasks, we validated its versatility and applicability as a general-purpose approach. Moreover, the proposed method is not restricted to the presented tasks and can be broadly applied. The motion prior used in our approach is trained on diverse human motion datasets, making it applicable not only to the racket movements shown in this paper but also to a wide range of downstream tasks. Furthermore, the inputs to both the task policy and the LMA consist of ego-centric representations of positions and velocities for each body part of the agent, the manipulated object, and the opponent. We consider this input representation to be generic and applicable to a wide range of tasks.

### 4.2 Training Procedure

To implement the proposed method, we pre-trained the motion prior $\pi_l$ and high-level policy $\pi_h$. As mentioned earlier, We used PULSE (Luo et al., 2024a) for training of the motion prior and high-level policy. In the pre-training stage, we utilized an off-the-shelf motion prior learned from the AMASS dataset (Mahmood et al., 2019). In the task stage, sufficient training time was allocated for both table tennis and tennis until the cumulative rewards converged, allowing the agents to acquire the skill to return the ball to specified positions. In the multi-agent interaction stage, we adopted self-play and trained both agent and opponent using alternating optimal strategies from Won et al. (2021).

Unlike the task stage, in the multi-agent interaction stage, the decision of where to return the ball is arbitrary. In this study, we interpret the target position of the ball included in $\mathbf{s}_t^g$ in the multi-agent interaction stage

as a self-determined target position and evaluated the proposed method in two scenarios: one where this value is set randomly and another where it is determined using the strategic-level network described later.

We executed 512 Isaac Gym environments in parallel on a Geforce RTX A6000, and both $\pi_h$ and $\pi_{LMA}$ were trained according to the rollouts collected using PPO (Schulman et al., 2017). We constructed both policies with a standard MLP, and the dimension of the latent motion $d_z$ was set to 32. $\pi_h$ had three hidden layers with dimensions of 2048, 1024, and 512, respectively. On the other hand, $\pi_{LMA}$ had two hidden layers with dimensions of 1024 and 512, respectively. As hyperparameters during training, we set $\gamma = 0.99$ and $\eta = 0.0001$. Additionally, the learning rate was set to $2 \times 10^{-5}$, and the frequency of agent swapping in self-play was set to once every 500 steps. Additional details on training procedures and hyperparameters are provided in Appendix A.2 and Appendix A.3, respectively.

## 4.3 Baselines

To verify the performance of the proposed method, we compared the following six baselines: (i) learning the high-level policy from scratch (PULSE (from scratch)), (ii) fine tuning of the high-level policy (PULSE (fine tuning)), (iii) fine tuning of the high-level policy with expanded input dimensionality (PULSE (finetuning w/ expansion)), (iv) residual RL for action-space correction (PULSE + residual RL (action) (Johannink et al., 2019)), (v) residual RL applied to the hidden layers of high-level policy (PULSE + residual RL (hidden-layer) (Perez et al., 2018)), and (vi) learning the high-level policy with KL prior (PULSE + KL (simplified (Liu et al., 2022))). Method (i) learns the high-level policy from scratch as prior studies. Method (ii) perform additional learning on the high-level policy acquired in single-agent tasks without $\mathbf{s}_t^{mi}$. Method (iii), like method (ii), fine-tunes the high-level policy, but additionally expands the input dimensionality so that the high-level policy can newly accept $\mathbf{s}_t^{mi}$. Specifically, $\mathbf{s}_t^{mi}$ is added to the input layer of the high-level policy $\pi_h$, and the weight matrix between the input layer and the first hidden layer is expanded by the dimensionality of $\mathbf{s}_t^{mi}$. The added weights are initialized randomly. Method (iv), following Johannink et al. (2019), learns a residual over the action space. It trains a residual policy $\pi_{res}$ that corrects the motion prior's output action $\mathbf{a}_t$ such that $\mathbf{a}_t' = \mathbf{a}_t + \pi_{res}(\mathbf{s}_t, \mathbf{s}_t^g, \mathbf{s}_t^{mi})$. Method (v), based on FiLM (Perez et al., 2018), adjusts the hidden-layer activations of the high-level policy. We train $\pi_{FiLM}(\mathbf{s}_t^{mi})$ to produce the correction terms $[\mathbf{c}_t^{(1)}, \cdots, \mathbf{c}_t^{(N)}, \mathbf{b}_t^{(1)}, \cdots, \mathbf{b}_t^{(N)}]$ for each hidden layer of the high-level policy. Let $\mathbf{x}_t^{(n)}$ denote the activation of the $n$-th hidden layer at time $t$. Its corrected value $\mathbf{x}_t'^{(n)}$ is given by $\mathbf{c}_t^{(n)}\mathbf{x}_t^{(n)} + \mathbf{b}_t^{(n)}$. Method (vi) is a simplified version of Liu et al. (2022). It adds a penalty based on KL divergence between the latent motion distributions output by the newly learned high-level policy $\pi_h'$ and the pre-trained high-level policy $\pi_h$. That is, we train $\pi_h'$ by maximizing the objective $\mathcal{J}_{mi} - w_{KL}D_{KL}[\pi_h'||\pi_h]$. Here, $D_{KL}$ denotes the KL divergence, and $w_{KL}$ is a hyperparameter that controls the strength of the KL penalty. In Liu et al. (2022), population-based training (Jaderberg et al., 2017) is also employed to simultaneously learn multiple hyper-parameters. However, this approach is computationally expensive and deviates from the rapid adaptation to multi-agent interactions. Therefore, it was not adopted in our experiments.

We evaluated the proposed method in two configurations: (vii) fixing the parameters of the high-level policy and training only the latent motion adjuster (PULSE + LMA (ours w/ freeze)) and (viii) simultaneously training both the high-level policy and the latent motion adjuster (PULSE + LMA (ours w/o freeze)).

As previously mentioned, in all methods, the ball's target position was determined randomly. The experiments that include learning the target position itself are presented in Appendix A.7.

## 5 Results

### 5.1 Cooperation in Multi-agent Interaction

Figure 3 shows the learning curve of each method for cooperative behavior learning in table tennis and tennis, indicating the average and standard deviation of the cumulative reward $\mathcal{J}$ over three trials. Figure 4 illustrates examples of the motions for each method at a specific moment during multi-agent cooperation. Additional results on the computational cost required for inference are provided in Appendix A.4 ; the effects of hyperparameters such as $\eta$, the number of network layers, and the length of the observation history are

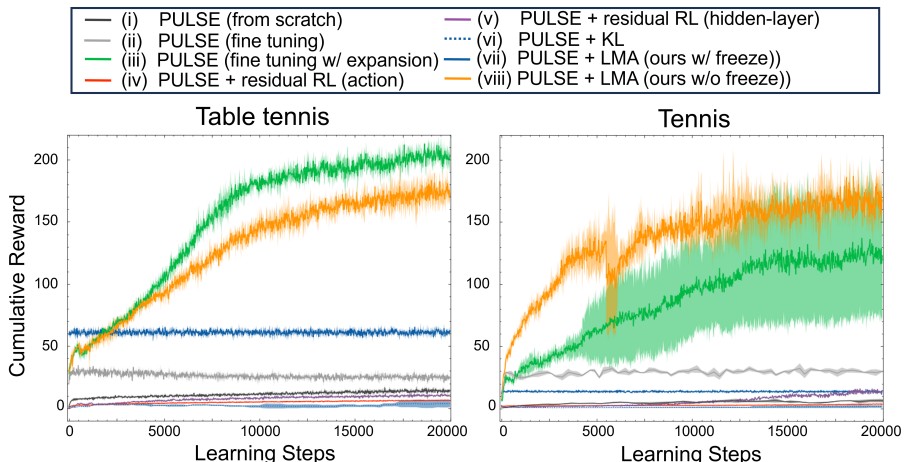

Figure 3: Comparison of learning performance with baseline methods in multi-agent cooperative tasks.

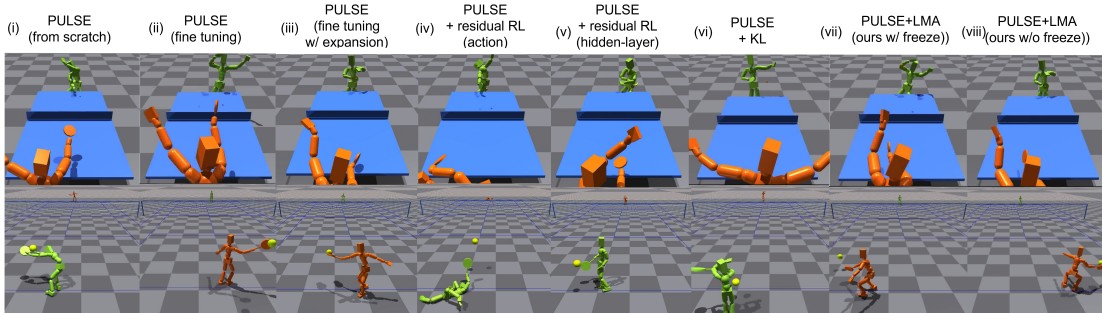

Figure 4: Example movements generated during cooperative multi-agent tasks (Green: Agent, Orange: Opponent). Top: Table tennis, Bottom: Tennis.

presented in Appendices A.5, A.6, and A.8, respectively; and visualizations of the latent space are included in Appendix A.9.

Method (i) and (vi) start by learning the basic motions required to hit the ball from scratch, which necessitates a significant amount of time for multi-agent interaction training. Consequently, within the range of 20,000 learning steps, they were unable to acquire adequate ball-hitting behaviors. Method (ii) resulted in low cumulative rewards because it returns the ball without considering the opponent's state. Methods (iv) and (v) apply residual reinforcement learning to adjust actions based on the opponent's state. However, modifying the action space itself or altering the hidden layers of the network risks significantly disrupting the skills already acquired. Consequently, these methods result in persistently low cumulative rewards. The proposed method (vii) shows high cumulative rewards in the very early stages of learning, demonstrating its ability to quickly adapt to the opponent's state. However, merely fine-tuning the latent motions is insufficient for acquiring advanced cooperative behaviors, leading to rapid convergence. On the other hand, the method (iii) and the proposed method (viii), which simultaneously learn the high-level policy, shows slower initial performance but achieves significantly high cumulative rewards in table tennis and tennis compared to other methods, enabling longer rallies. This is likely due to its ability to perform flexible motion adjustments that cannot be addressed through fine-tuning alone.

Interestingly, although method (iii) exhibits higher learning performance than the proposed method (viii) in the table-tennis task, its learning becomes highly unstable in the tennis task. This discrepancy is likely attributable to two factors: the difference in the types of input information emphasized by each method,

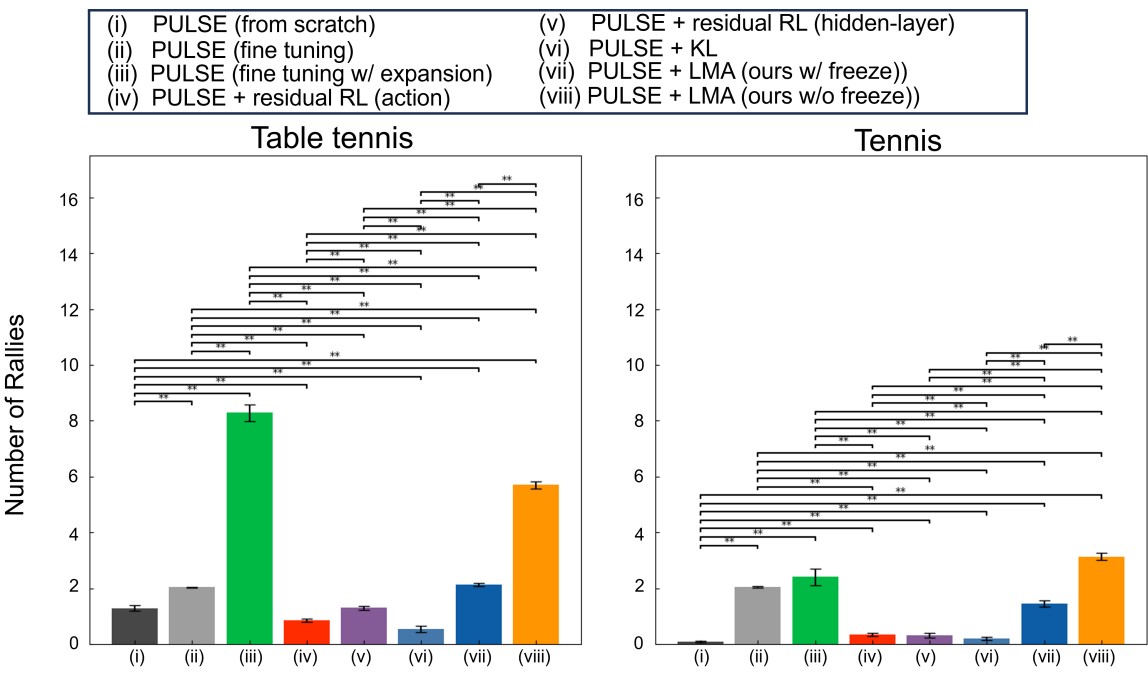

Figure 5: The average number of rallies for each method (*: $p < 0.05$, **: $p < 0.01$).

and the difference in which agent plays a critical role in coordination across tasks. In method (iii), both the task-dependent state $\mathbf{s}_t^g$ and the opponent state $\mathbf{s}_t^{mi}$ are simultaneously fed into the high-level policy $\pi_h$, and because these two sources of information are compared relative to each other, the value of the opponent-state information tends to be underestimated (see Appendix A.10 for details). In contrast, as demonstrated by the sensitivity analysis in the next section, the proposed method (viii) selectively inputs the opponent state only into the LMA, resulting in a functional division in which the LMA responds primarily to the opponent state while the high-level policy responds primarily to the task state. From the perspective of task characteristics, in the table-tennis task — where lower-body movement is almost unnecessary — the receiving agent can adapt to diverse ball trajectories simply by learning appropriate hitting motions. In tennis, however, whole-body control involving locomotion is required, making it difficult for the receiving agent alone to handle all possible ball trajectories. Consequently, it becomes crucial for the hitting agent to adjust the ball trajectory. Taken together, method (iii) performs well in tasks like table tennis, where receiving motions based on task-state information are essential, but performs poorly in tasks like tennis, where hitting motions based on opponent-state information play a more critical role. In contrast, the proposed method enables robust learning across both tasks without sacrificing stability, because the LMA and the high-level policy divide their responsibilities in a way that naturally accommodates differences in the types of information required for coordinated behavior.

Figure 5 shows the mean and standard deviation of the rally counts based on 10000 rally trial using the models of each method after 20,000 learning steps. We extracted 100 mean values of rally counts obtained from 100 rally trials each, thereby transforming them to follow a normal distribution under the central limit theorem. We assessed group differences using Welch's ANOVA (Welch, 1951). As the result was significant ($p < 0.01$), post hoc comparisons were conducted using the Games–Howell test (Games & Howell, 1976). * and ** indicate statistical significance at $p < 0.05$ and $p < 0.01$, respectively. Consistent with the results in Fig. 3, method (iii) achieves significantly higher rally counts than the other methods in the table-tennis task, whereas the proposed method (viii) outperforms all others in the tennis task.

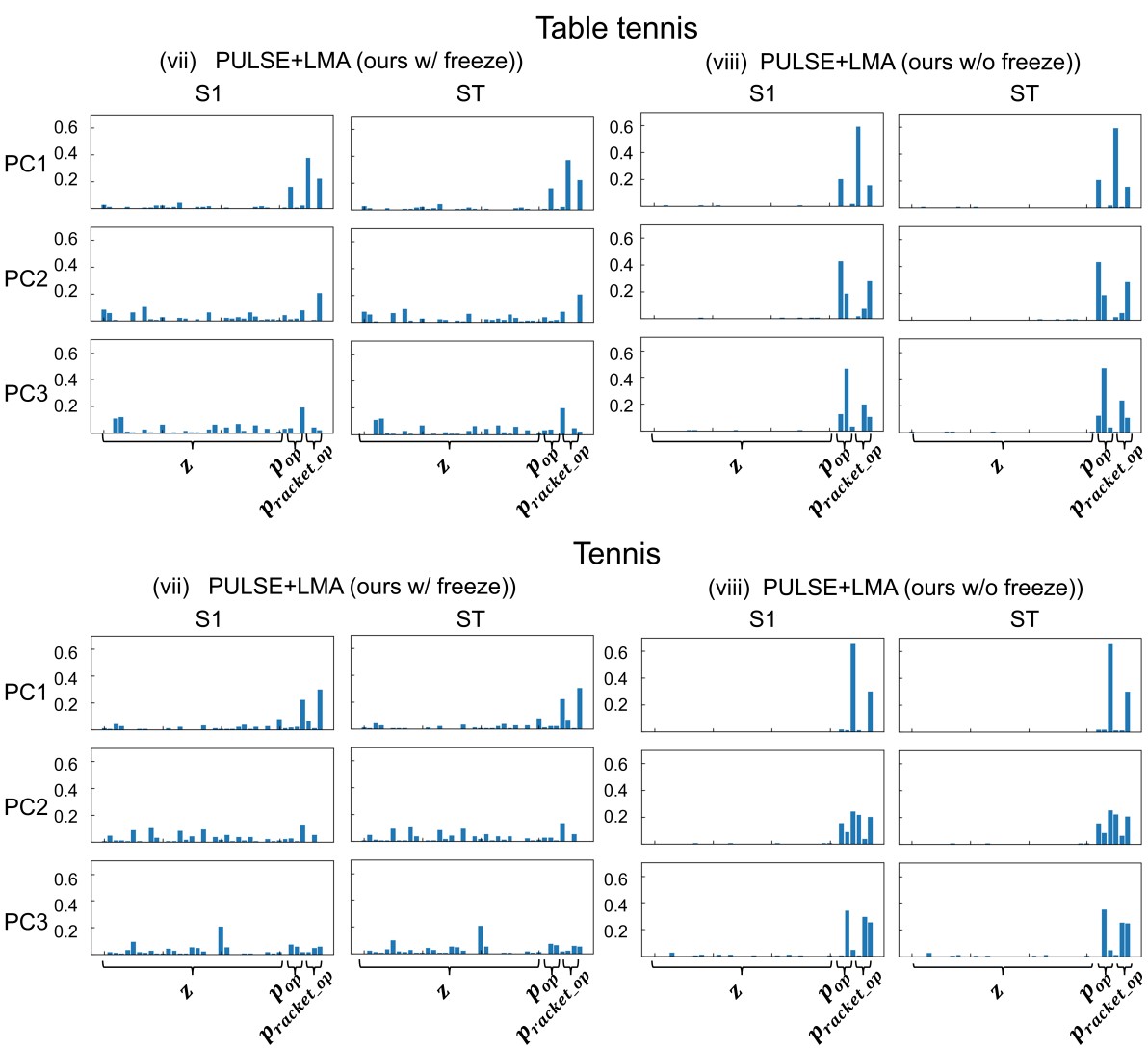

Figure 6: Sobol sensitivity analysis results for LMA (top: table tennis, bottom: tennis. left: method (v), right: method (vi)).

**Sensitivity of LMA to Observations**   To verify whether the LMA can adjust the latent motion according to the observed opponent state, we conducted a Sobol sensitivity analysis (Sobol′, 2001; Saltelli, 2002), a method that quantifies how each input dimension contributes to the variability of a model's output by decomposing the output variance into components attributable to each input. Using 10,000 observation samples obtained from the proposed methods (vii) and (viii), we computed the range of possible values for each input dimension of the LMA. Within these ranges, we generated $1000 \times (2 \times 38 + 2) = 78,000$ samples using the Saltelli sampler. We then applied PCA to $\Delta z$, the output of LMA, and measured the contribution of each input dimension to the variance of each principal component.

Figure 6 presents the results of the Sobol sensitivity analysis for LMA in both table tennis and tennis. The left column corresponds to method (vii), and the right column corresponds to method (viii). S1 denotes the first-order indices, and ST represents the total-order indices. From top to bottom, bar charts show the sensitivity indices for the first to third principal components of $\Delta z$. The horizontal axis of each bar chart represents the input dimensions of LMA, ordered as follows: the dimensions of the latent motion $z$ output

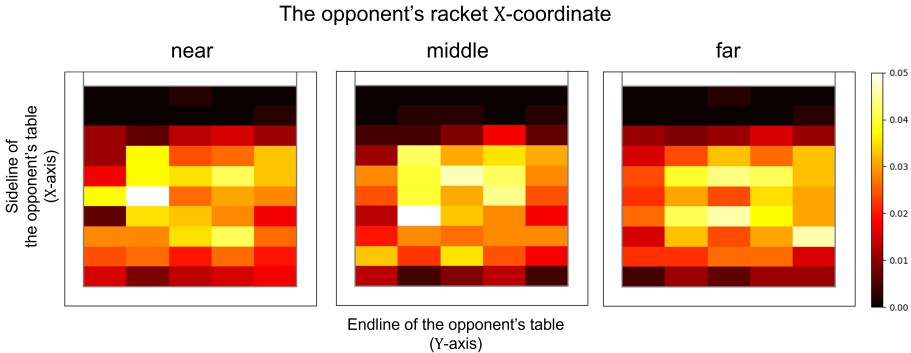

Figure 7: Ball return position visualization for method (viii) in the table tennis rally task

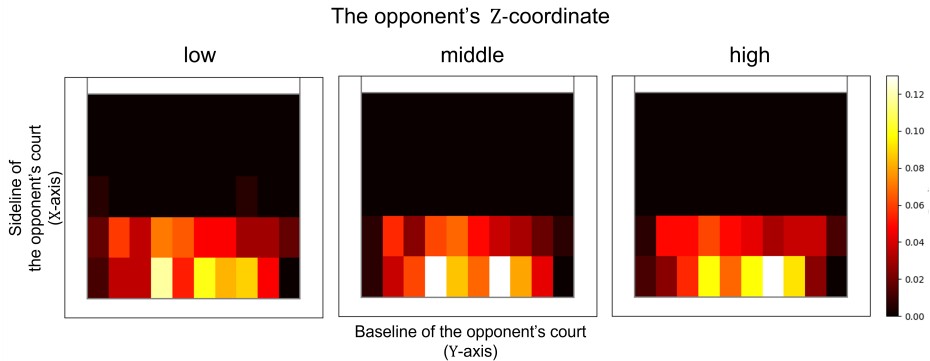

Figure 8: Ball return position visualization for method (viii) in the tennis rally Task

by the high-level policy, the $x$, $y$, and $z$ components of the opponent's position $\mathbf{p}_{op}$, and the $x$, $y$, and $z$ components of the opponent's racket position $\mathbf{p}_{racket\_op}$. Here, the $x$-axis is aligned with the sideline, while the $y$-axis is parallel to the endline or baseline. Method (vii), which uses a fixed high-level policy, allows many input dimensions to influence the output of the LMA, whereas method (viii), which jointly trains the high-level policy, results in the LMA being affected only by the opponent's state.

From the results of method (viii) in Figure 6, we can also observe that the LMA adjusts the latent motion by selectively focusing on different opponent-related input dimensions depending on the environment. In table tennis, the opponent's racket position in the $x$-direction is the most influential factor, whereas in tennis, the agent's behavior varies according to the opponent's $z$-coordinate. Figure 7 visualizes the locations on the table where the agent returned the ball under method (viii), conditioned on the opponent's racket $x$-coordinate. The figure provides a top-down view of the opponent's side of the table, where the opponent's racket $x$-coordinate at the moment the agent hits the ball is grouped into three categories (near: 2.5 m, medium: 3.0 m, and far: 3.5 m). The colors represent the return probability for each region, with brighter areas indicating higher frequencies of ball bounces under that condition. It shows that the agent appropriately adjusts its behavior to ensure that the ball bounces at a suitable height near the opponent's hitting point: when the opponent's racket is close, the ball is returned to bounce near the center of the table, whereas when the racket is farther away, the ball is returned to bounce closer to the table's edge. Similar to Figure 7, Figure 8 visualizes how the ball's landing position on the opponent's court changes under method (viii) when the opponent's $z$-coordinate is grouped into three categories (low: $-0.3$ m, medium: 0.0 m, and high: 0.3 m). In the single-agent setting, the learned behaviors primarily consist of low-center-of-gravity backhand shots and high-center-of-gravity forehand shots. Figure 8 suggests that the agent exploits this tendency: when the opponent's center of gravity is low, the agent returns the ball toward the left side of the court, where backhand shots are easier to execute, whereas when the opponent's center of gravity is high, the agent directs the ball toward the right side, facilitating forehand shots.

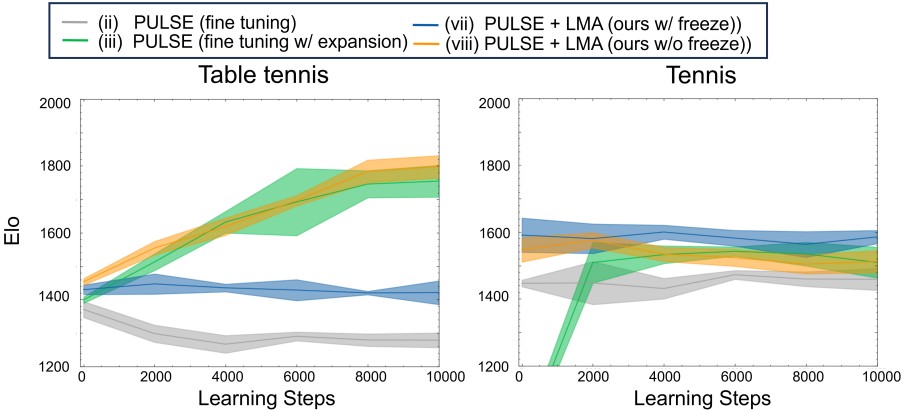

Figure 9: Elo score of each method in multi-agent competitive tasks.

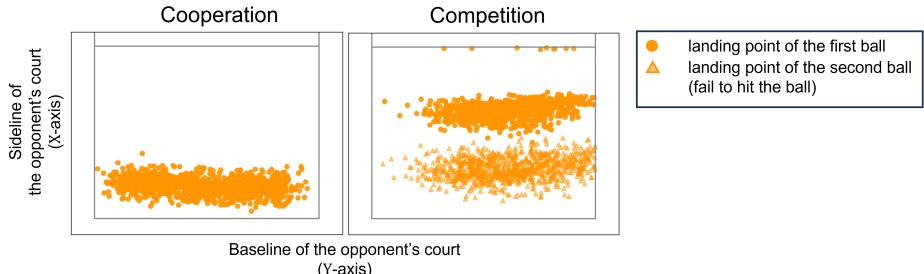

Figure 10: Ball landing positions on the opponent's court when using method (viii) for multi-agent interactions in tennis.

## 5.2 Competition in Multi-agent Tennis Matches

Figure 9 shows the transitions of Elo scores (Elo, 1967) for each method in the competitive tasks for table tennis and tennis. The Elo score is an indicator of an agent's strength and is commonly used for evaluating competitive outcomes. We dumped the weights of each method every 2000 learning steps during the training process of the competitive tasks in table and tennis across three trials. The results of methods (i), (iv), (v) and (vi) were excluded as they did not sufficiently acquire ball-hitting motions. Consequently, a total of 72 models were prepared by dumping weights six times during training for each of the four remaining methods across three trials. We conducted a round-robin tournament using these models, where each pair of models played 100 matches. Elo scores were calculated based on the win-loss results for each model combination. In the table tennis task, it can be observed that the method (iii) and the proposed method (viii) become significantly stronger as training progresses. This indicates that, as in the cooperative setting, both the additional training of the high-level policy with expanded input dimensions and the proposed method were able to acquire sophisticated competitive behaviors. Unlike the table tennis task, in the tennis tasks, the transitions of Elo scores do not improve across all methods, including the proposed method (viii). This is due to the insufficient performance of the policies acquired in the single-agent stage, resulting in certain strategies that the agents are unable to adapt to. In the single-agent task, the motion of returning the ball from in front of the endline is rarely learned. Therefore, the proposed method acquires an adversarial policy that returns the ball to a position closer to the net, which is more difficult for the opponent.

Figure 10 shows the landing positions of the ball on the opponent's court when returned using the proposed method (viii) in both the tennis rally task and the tennis match task. The first ball's landing point is indicated by a circle, and the second ball's landing point is indicated by a triangle. In the cooperative task of the tennis rally, the ball is returned to bounce once near the back of the opponent's court so that it rebounds

towards the opponent's racket. However, in the competitive task, many trajectories are aimed to bounce once near the net. As a result, the ball often lands twice within the court before the opponent can return it. To acquire more advanced competitive motions, it is desirable to perform additional training in situations where the agent was unable to respond effectively. We plan to develop a continuous skill improvement framework that relearns the weaknesses of agents identified in multi-agent interactions. Furthermore, although the proposed method can adapt its actions based on the opponent's immediate states, it is unable to adjust its own behavior through opponent-strategy inference, as commonly employed in multi-agent reinforcement learning (Xie et al., 2021; Wang et al., 2022). Consequently, it is difficult for the method to acquire globally optimal behaviors, such as anticipating a drop shot and moving forward preemptively. Developing advanced strategic interactions based on opponent strategy inference remains a challenge for future work.

## 6 Conclusion and Future Works

In this paper, we propose a new method that utilize both motion prior and single-agent task policy for multi-agent interaction. By learning a latent motion adjuster that adjusts the latent motion output by the single-agent task policy based on the opponent's state, it becomes possible to utilize the skills acquired in the single-agent setting while outputting appropriate actions according to the opponent's state. Through simulated sport tasks, we verified that the proposed method achieved higher cooperative and competitive motions at a comparable or faster learning speed than conventional methods. Applying this method to diverse multi-agent interaction tasks that require advanced strategies, including making significant changes to action selection based on the opponent's intentions, remains a challenge for future work. We also plan to explore applications beyond ball-related tasks, including collaborative robot manipulation.

### Acknowledgments

This work was supported by JSPS KAKENHI Grant Number JP22H04998 and JST ASPIRE Grant Number JPMJAP2503.

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

# A  Appendix

## A.1  Experimental Details

We used simulated table tennis and tennis environments in SMPLOlympics (Luo et al., 2024b). These environments utilize a humanoid with the kinematic structure of SMPL (Loper et al., 2015), which has 24 joints, 23 of which are controllable. The control cycle for the humanoid output by the proposed method and baseline methods was set to 30 Hz.

In the table tennis environment, a circular racket with an 8 cm radius is attached 12 cm away from the humanoid's right wrist (Left of Fig. 11). The dimensions of the table and ball conform to standard sizes. The table is 1.525 m wide, 2.74 m long, and 0.76 m tall, with a net height of 15.25 cm, and thw ball has a radius of 2cm.

In the tennis environment, a circular racket with a radius of 15 cm is attached 35 cm away from the humanoid's right wrist (Right of Fig. 11). The dimensions of the court are the same as actual tennis courts, with a width of 8.23 m, a length of 23.77 m. Additionally, the net height is 1 m, ant the ball has a radius of 3.2 cm.

In both environments, at the start of each episode, a ball is launched into the player's court with a random speed and direction. The target position for the ball's landing point $\mathbf{p}^*_{ball}$ is randomly assigned within the opponent's court at the start of each episode or when the opponent returns the ball. In methods that use the strategic-level network described in the main text, this target position is replaced by its output. Additionally, at each time step, the predicted arrival point of the ball at the end line of the agent's court is calculated based on the current ball position and velocity. The agent's target position $\mathbf{p}^*_{self}$ is then set such that the racket aligns with this predicted arrival point. Each episode ends when the ball is missed or goes out of bounds and the outcome of each episode is also determined based on these results in competitive tasks.

## A.2  Reward Design and Training Details

In the task stage for both table tennis and tennis, we used task-dependent reward $r^g_t$ designed in SMPLOlympics (Luo et al., 2024b).

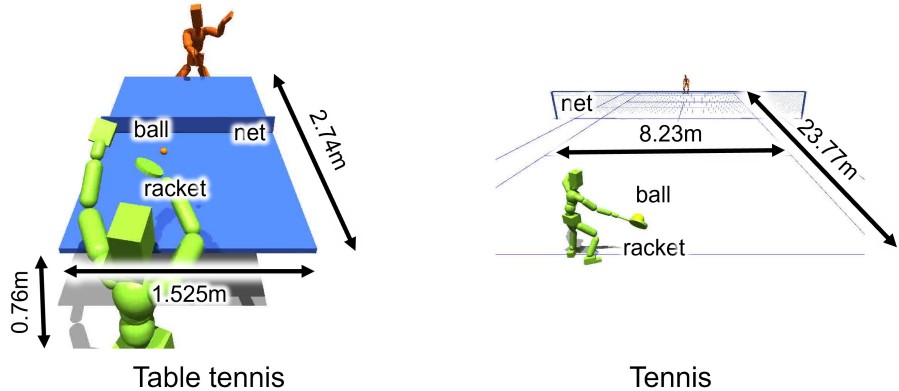

Figure 11: Components and sizes of table tennis and tennis.

In table tennis task, we used the following reward,

$$
\begin{aligned}
r_t^g =& \mathbb{I}(N_c = 0) \times e^{-4d_{br}^2 - 4d_{st}^2 - n_z} + \\
& \mathbb{I}(N_c > 0) \times e^{-4d_{st}^2}(e^{-4d_{lt}^2} + 2 + N_o) + \\
& \mathbb{I}((N_c > 0) \wedge (d_{lt} < 0.01)) \times e^{-4d_{st}^2},
\end{aligned}
\tag{5}
$$

where $d_{br}$, $d_{st}$, and $d_{lt}$ is the distance between the ball and the racket, the current agent position and the agent's target position $\mathbf{p}_{self}^*$, and the ball's landing position and the target ball position $\mathbf{p}_{ball}^*$, respectively. $n_z$ denotes the z-direction value of the racket's normal. $N_c$, $N_l$ represent the number of times the ball makes contact with the racket and lands on the court, respectively. $N_c$ and $N_l$ are reset to 0 each time the opponent returns the ball in the multi-agent interaction stage. The first term on the right-hand side of Eq. 5 rewards the agent for adopting a position and posture that makes it easier to return the ball. The second and third term encourage the agent to maintain its position while returning the ball to the target location.

In the tennis task, we used the following reward

$$
\begin{aligned}
r_t^g =& \mathbb{I}(N_c = 0) \times e^{-d_{br}^2 - 2d_{st}^2} + \\
& \mathbb{I}((N_c > 0) \wedge (z_b < 3)) \times e^{-d_{lt}^2}(e^{-2d_{st}^2} + 0.1) + \\
& \mathbb{I}((N_l > 0) \wedge (z_b < 3)) \times (e^{-2d_{st}^2} + 0.1) + \\
& \mathbb{I}((N_l > 0) \wedge (d_{lt} < 0.1)) \times (e^{-2d_{st}^2} + 0.1),
\end{aligned}
\tag{6}
$$

where $z_b$ is the height of the ball.

In the task stage for tennis, we utilized $r^{AMP}$, the reward from AMP (Peng et al., 2021), in addition to the task-dependent rewards. $r^{AMP}$ reflects the difficulty of distinguishing between motions sampled from the motion dataset and those actually obtained in the simulation, thereby promoting the acquisition of natural motions. We calculated $r^{AMP}$ based on the tennis motion data provided by SMPLOlympics and set the overall reward given during the task stage as $0.5r_t^g + 0.5r_t^{AMP}$. On the other hand, we observed that using $r^{AMP}$ in table tennis significantly reduced performance. This is likely because the provided data consists of motion extracted from videos, and for table tennis, the extraction quality leads to the inclusion of physically inappropriate movements. Therefore, in the task stage for table tennis, we trained the high-level policy using only $r_t^g$.

As mentioned in the main text, the performance of the high-level policy significantly influences the subsequent performance in multi-agent interactions. Therefore, in the task-stage, training was continued for sufficient time even after the learning curve had approximately converged, and the final weights were used for multi-agent interactions. Specifically, 70,000 steps of training were conducted for table tennis and 180,000 steps

Table 1: Hyperparameters for each method

| Method | Hyperparameter | Value |
|---|---|---|
| PULSE (Luo et al., 2024a) | $\pi_l$ hidden layer | [3096, 2048, 1024] |
| | $\pi_h$ hidden layer | [2048, 1024, 512] |
| | activation | SiLU |
| | learning rate | 2e-5 |
| | $\alpha$ | 0.005 |
| | $\beta$ | 0.01 ($2.5 \times 10^9$ samples) $\rightarrow$ 0.001 ($5 \times 10^9$ samples) |
| | $\gamma$ | 0.99 |
| | latent motion dimension $d_z$ | 32 |
| LMA | $\pi_{LMA}$ hidden layer | [1024, 512] |
| | activation | SiLU |
| | $\eta$ | 0.0001 |
| residual RL (action) (Johannink et al., 2019) | $\pi_{res}$ hidden layer | [1024, 512] |
| | activation | SiLU |
| residual RL (hidden layer) (Perez et al., 2018) | $\pi_{FiLM}$ hidden layer | [1024, 512] |
| | activation | SiLU |
| KL prior (Liu et al., 2022) | $w_{KL}$ | 0.5 |

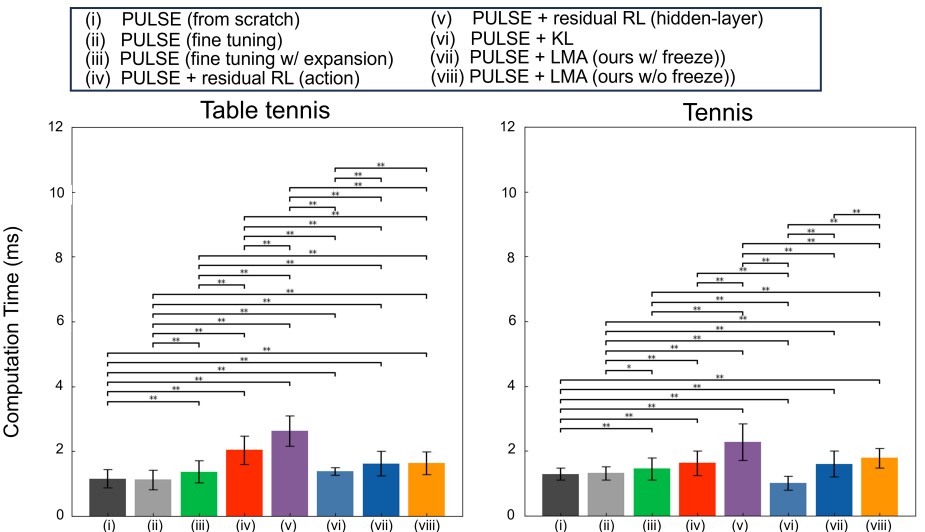

Figure 12: Results of the computation time required for a single action output across each method (*: $p < 0.05$, **: $p < 0.01$).

for tennis. When running 2,048 Isaac Gym environments in parallel on a Geforce RTX A6000, the training took approximately one to two days.

## A.3 Hyperparameters for Each Method

Table 1 summarizes the hyperparameters used for each method. Hyperparameters not explicitly specified for a given method were set to the same values as those used in PULSE.

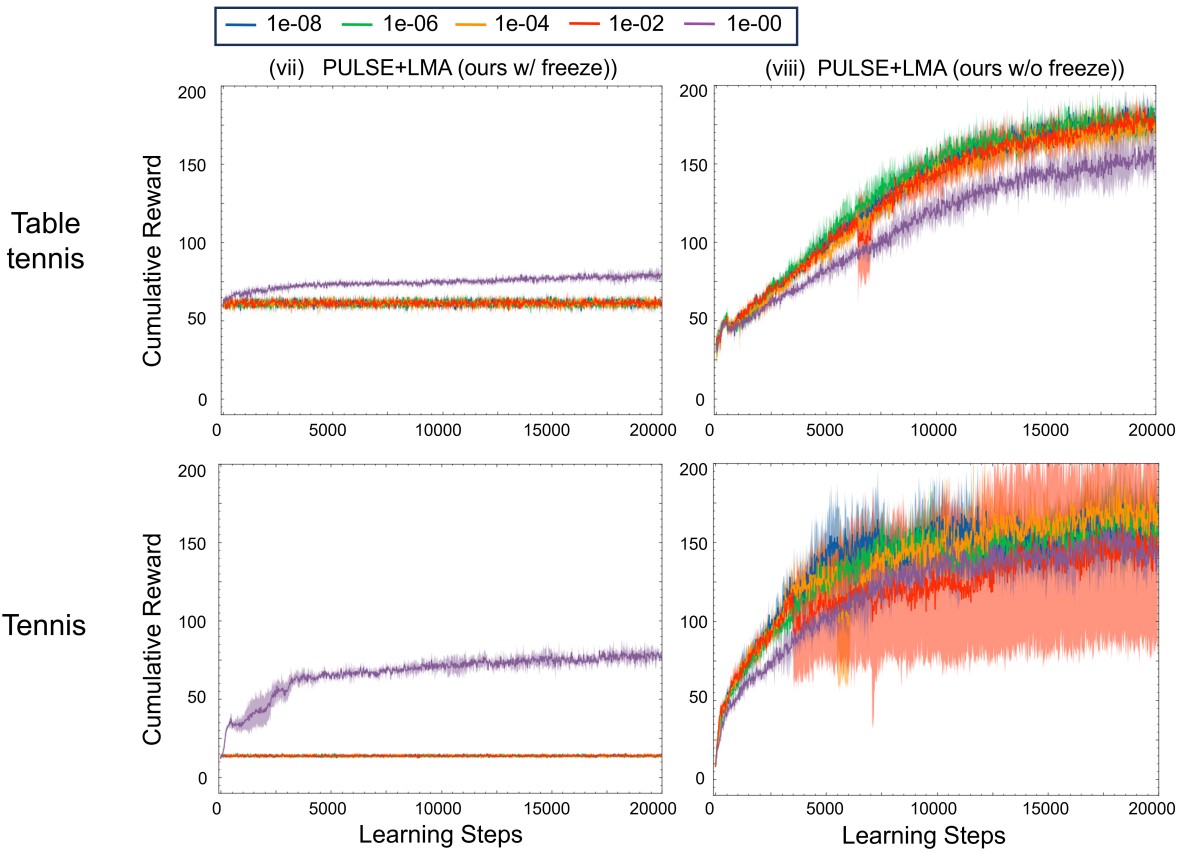

Figure 13: Results of the learning curves obtained for each value of $\eta$ (left: method (vii), right: method (viii))

.

## A.4 Computational Costs

Figure 12 shows the results of comparing the computation time required for a single action output across each method. We extracted 100 mean values of computation times obtained from 10000 samples, thereby transforming them to follow a normal distribution under the central limit theorem. We assessed group differences using Welch's ANOVA (Welch, 1951). As the result was significant ($p < 0.01$), post hoc comparisons were conducted using the Games–Howell test (Games & Howell, 1976). * and ** indicate statistical significance at $p < 0.05$ and $p < 0.01$, respectively. Compared with the conventional method that uses only the high-level policy, the proposed methods require longer computation times. However, both the task policy and the LMA are lightweight two-layer MLPs, which are faster to compute than the simulation update steps. Therefore, the total computational cost required for training remains largely unchanged regardless of whether the LMA is incorporated or the task policy is trained jointly.

## A.5 Effect of the Adjustment Scale in LMA

To assess the influence of $\eta$, which determines the adjustment scale in LMA, we conducted three training runs for each $\eta$ value ([1e−8, 1e−6, 1e−4, 1e−2, 1e−0]) in the proposed methods (vii) and (viii), and evaluated the resulting learning curves.

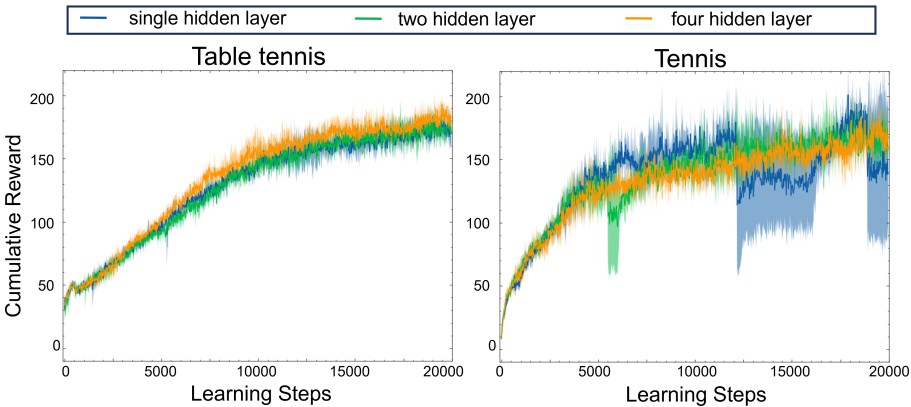

Figure 14: Learning curves of method (viii) across different numbers of LMA network layers.

Figure 13 shows the results of the learning curves obtained for each value of $\eta$. In method (vii), $\eta = 1$ yields higher learning performance than the other values, whereas in method (viii), $\eta = 1$ results in lower performance. This suggests that the role of LMA differs depending on whether the high-level policy is trained simultaneously. Under the experimental setting of this study, it is difficult to achieve efficient cooperative behavior using only the high-level policy trained in a single-agent task. In method (vii), which uses a fixed high-level policy, LMA functions as a residual reinforcement learning module that compensates for the high-level policy; therefore, setting an adjustment scale comparable to that of the high-level policy is important for effective learning. In contrast, in method (viii), where the high-level policy is trained jointly, LMA serves as an auxiliary component that provides fine-grained adjustments based on the opponent's state. Consequently, assigning an adjustment scale comparable to that of the high-level policy can lead to interference, ultimately degrading performance.

## A.6 Effect of the Number of LMA Network Layers on Performance

In this section, we conduct an ablation study to evaluate how the number of network layers in the lightweight LMA architecture affects its performance. We compare three network configurations with 1, 2, and 4 hidden layers. The dimensionalities of the hidden layers are set to [1024] for the single-layer network, [1024, 512] for the two-layer network, and [1024, 512, 512, 256] for the four-layer network.

Figure 14 presents a comparison of the learning curves for different numbers of network layers in LMA. When the high-level policy is trained jointly with LMA (Method (viii)), the additional learning of the high-level policy contributes most to the performance, and thus the differences caused by the number of LMA network layers are minimal.

## A.7 Combination with a Strategic-level Network

Although the proposed method enables fine-grained adjustments of latent motions, it is difficult to substantially modify the strategy itself — for example, making large changes to the ball return position. In Section 4, we evaluated multi-agent interactions in a setting where the target landing position of the ball was randomly specified, but ideally, this target position should also be adaptable based on the opponent's state. Therefore, we conducted an evaluation combining the strategic-level network proposed by Han et al. (2024). The strategic network $\pi_{strategy}$ has the same architecture as the high-level policy $\pi_h$, and its output is directly used as the target landing position.

To evaluate the effect of LMA when introducing the strategic network, we compare the following three methods. Method (ix) applies PULSE to Han et al. (2024) and learns a strategic-level policy $\pi_{strategy}$ (PULSE + SEPMC (Han et al., 2024)). Method (x) corresponds to our proposed method, where $\pi_h$ is frozen while $\pi_{LMA}$ and $\pi_{strategy}$ are trained jointly (PULSE + LMA + SEPMC (ours w/ freeze)). Method (xi)

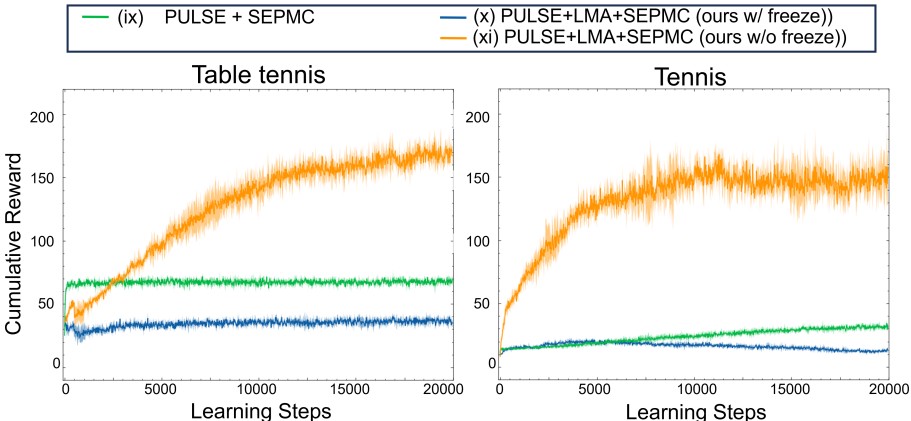

Figure 15: Comparison of learning performance with baseline methods when using the strategy-level network in multi-agent cooperative tasks.

corresponds to our proposed method, where $\pi_h$, $\pi_{LMA}$ and $\pi_{strategy}$ are all trained jointly (PULSE + LMA + SEPMC (ours w/o freeze)).

Figure 15 shows the learning curves of each method for cooperative behavior learning in the table-tennis and tennis environments when using the strategy-level network, presenting the mean and standard deviation of the cumulative reward $J$ over three trials. In method (ix), which augments the high-level policy only with the strategy-level network, the cumulative reward remains low. This is likely because, under the setting where only the ball's target position is learned, it becomes difficult to finely adjust behaviors according to the opponent's state. Method (x) also fails to acquire sophisticated cooperative behaviors; even with additional strategy learning, merely fine-tuning the latent motion is insufficient. In contrast, method (xi) achieves high cumulative rewards by jointly learning the high-level policy, suggesting that it enables more flexible motion adaptation. However, the learning curves of method (viii) in Fig. 3 and method (xi) in Fig. 15 are nearly identical, indicating that the strategy-level network contributes little to overall performance. This is likely because the skills acquired in the single-agent stage are robust to movements along the baseline, reducing the necessity to adjust target positions.

## A.8 Effect of Observation-History Length on Performance

In general, using not only the current observation of the opponent's state but also its history is effective for predicting the opponent's strategy. Therefore, we additionally evaluated whether the cooperation performance of LMA improves when it is provided with a history of the opponent's states. For comparison, we assessed three settings in which the length of the opponent-state history given to LMA was set to 1 (i.e., only the current opponent state), 4, and 8, respectively.

Figure 16 compares the learning curves obtained when varying the length of the opponent-state history in the multi-agent cooperation task. Each curve shows the mean and standard deviation of the cumulative return $J$ over three runs. As shown in Fig. 16, contrary to expectation, the length of the opponent-state history has little effect on performance in the cooperative tasks for table tennis and tennis considered in this study. One possible reason is that, in the table-tennis and tennis environments used here, interactions between the ball and the agents occur instantaneously. That is, the agents' positions matter only at the moment when either player hits the ball, and at all other times the opponent's state does not influence the ball's trajectory. Consequently, effective cooperative behavior can be acquired even from purely instantaneous information.

A promising direction for future work is to apply the proposed method to tasks in which an opponent's intent prediction is essential, such as navigating through a crowded environment. This would allow us to evaluate whether the method can also handle long-horizon strategic reasoning.

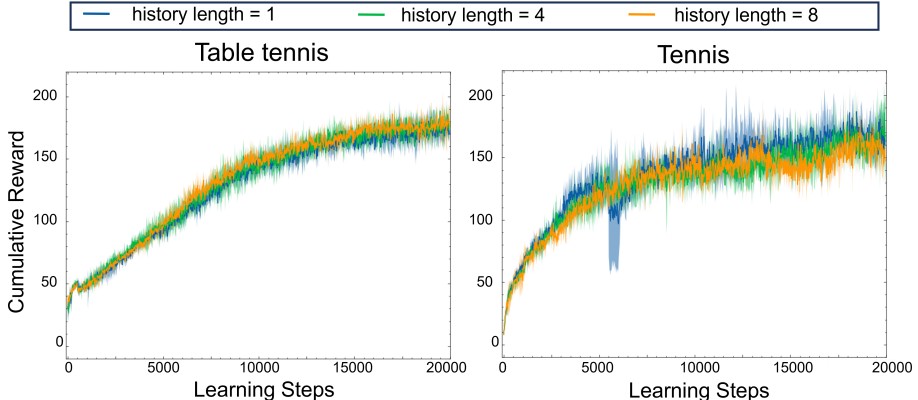

Figure 16: Comparison of learning curves with different lengths of opponent-state observation histories.

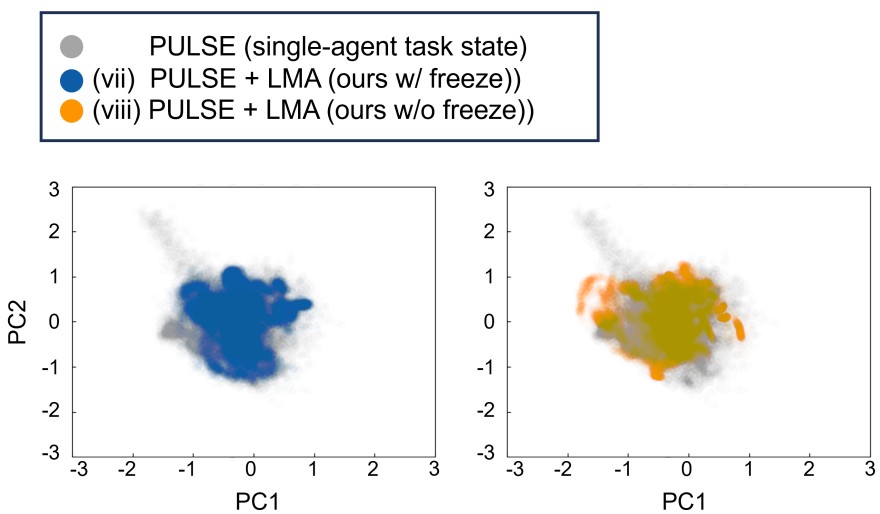

Figure 17: Latent motion space visualization for proposed methods.

## A.9 Visualization of Latent Motions

Figure 17 visualizes the latent motion spaces obtained from the high-level policy learned in the single-agent task stage, as well as those obtained by method (iii) and the proposed methods (vii) and (viii) in the multi-agent interaction stage. We applied PCA to 10,000 latent motions sampled from the high-level policy trained in the single-agent task stage, and overlaid the projections of 10,000 latent motions sampled from each method onto the first and second principal components. A darker color indicates a higher sampling frequency. From Fig. 17, we observe that each method learns latent motions suitable for multi-agent interaction without deviating substantially from the latent motion space acquired in the single-agent task stage. This suggests that leveraging the skills learned in the single-agent setting is beneficial for multi-agent interaction.

Figure 18 visualizes the shift in latent motions induced by LMA. The horizontal and vertical axes correspond to the first and second principal components used in Fig. 17, and it illustrates the transitions of latent motions over 30 steps starting from a given time step. The gray solid line represents the transitions of the latent motions before adjustment by LMA, while the dashed line represents the transitions after adjustment by LMA. From Fig. 18, we observe that the magnitude of the adjustments made by LMA is sufficiently small compared to the transition magnitude of the latent motions at each time step. This indicates that LMA leverages the latent motions output by the high-level policy while performing fine-grained adjustments.

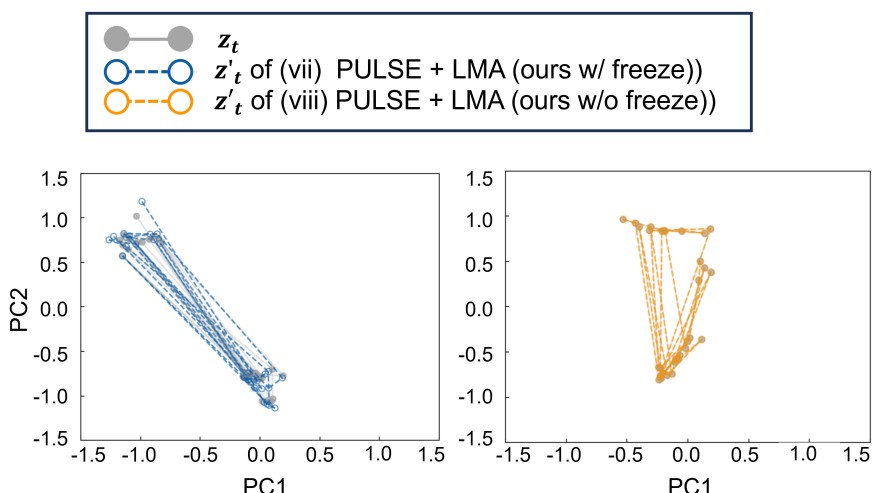

Figure 18: Latent motion transition visualization for proposed methods.

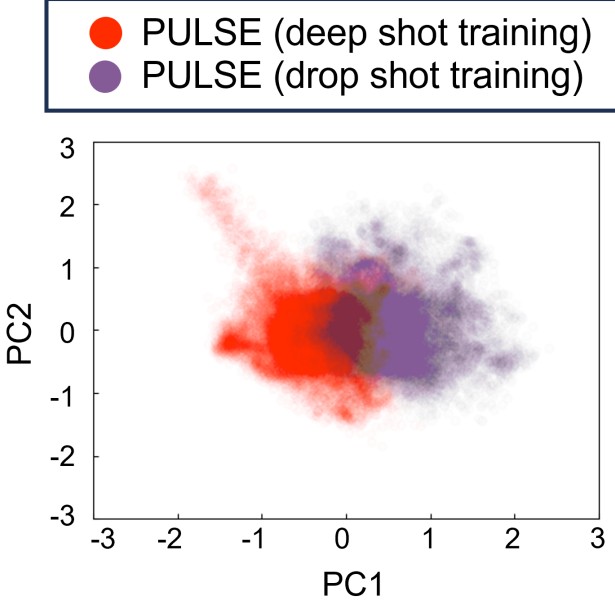

Figure 19: Differences in latent motion coverage across training conditions.

While the proposed method has been shown to effectively leverage the latent motions produced by the high-level policy acquired in the single-agent task, this also implies that appropriate multi-agent interaction may become difficult if the required behaviors are not covered within this latent motion range. Figure 19 illustrates the coverage of latent motions when the agent is trained under different conditions in the task stage of tennis. The red region represents the latent motions obtained when learning return behaviors for ball trajectories that bounce near the baseline, such as deep shots. The purple region represents the latent motions obtained when learning return behaviors for ball trajectories that bounce near the net, such as drop shots. Although some overlap exists between the two conditions, Fig. 19 shows that different latent motions are required depending on the condition. Therefore, as described in Section 5.2, an agent trained only on deep shots struggles to immediately respond when encountering an unseen drop shot during multi-agent

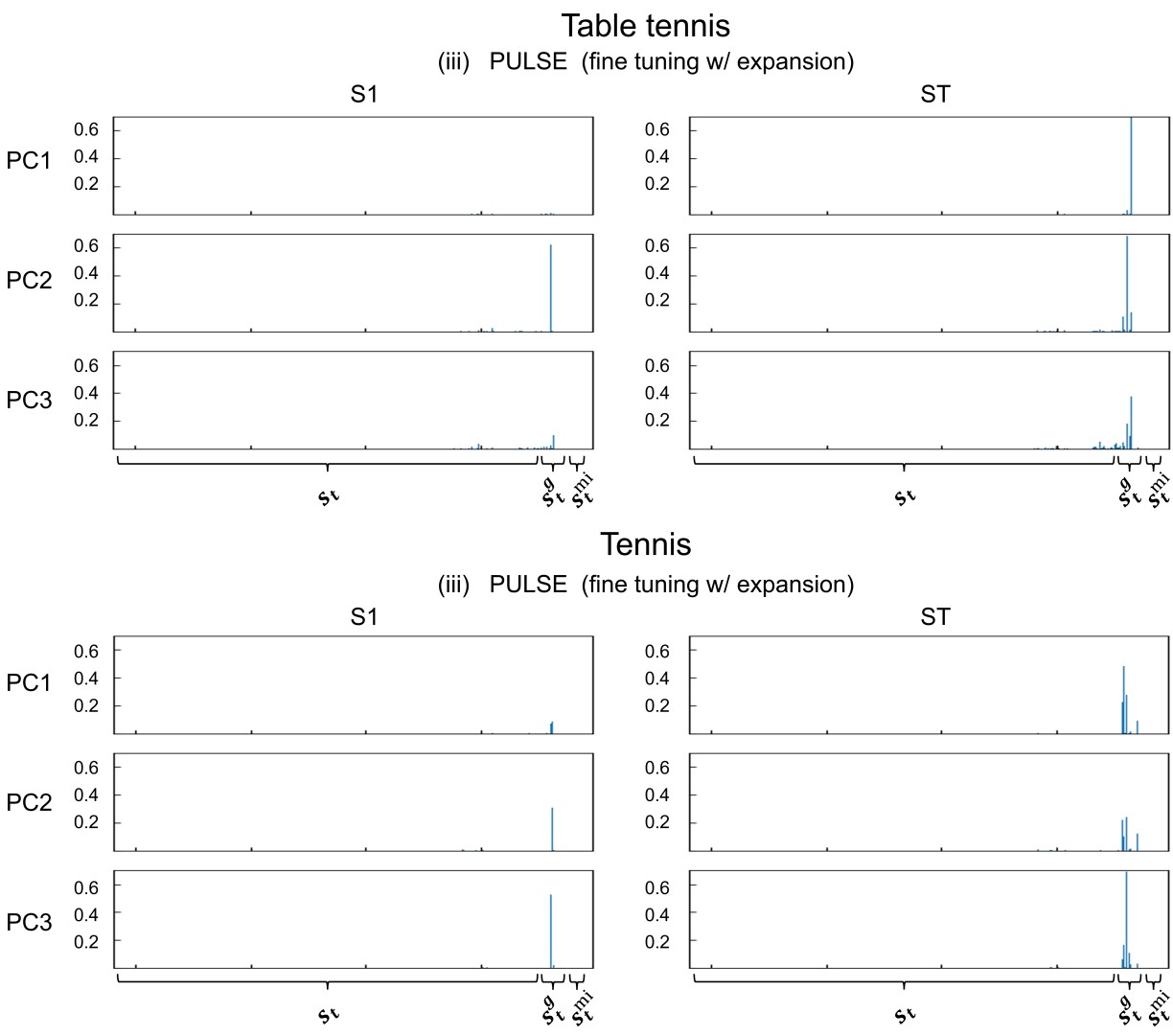

Figure 20: Sobol sensitivity analysis results for fine tuning with input-dimension expansion (method (iii)).

interaction. To address this issue, a framework for continual skill improvement, as discussed in Section 5.2, will be necessary in future work.

## A.10 Sensitivity of Fine-Tuning with Input-Dimension Expansion to Observations

To examine which input information method (iii) — that is, the high-level policy fine-tuned after expanding its input dimensionality to newly accept $\mathbf{s}_t^{mi}$ — actually responds to, we conducted a Sobol sensitivity analysis (Sobolʹ, 2001; Saltelli, 2002). Using 10,000 observation samples obtained from method (iii) and (viii), we computed the range of possible values for each input dimension of the high-level policy. Within these ranges, we generated $1000 \times (2 \times 379 + 2) = 760,000$ samples using the Saltelli sampler. We then applied PCA to $z$, the output of high-level policy, and measured the contribution of each input dimension to the variance of each principal component.

Figure 20 presents the results of the Sobol sensitivity analysis for method (iii) in both table tennis and tennis. S1 denotes the first-order indices, and ST represents the total-order indices. From top to bottom,

bar charts show the sensitivity indices for the first to third principal components of $z$. The horizontal axis of each bar chart represents the input dimensions of high-level policy, ordered as follows: the dimensions of the agent's proprioceptive state $\mathbf{s}_t$, the task-dependent state $\mathbf{s}_t^g$ and the opponent state $\mathbf{s}_t^{mi}$. From Fig. 20, we observe that in method (iii), task-dependent states exert a much stronger influence on the output of the high-level policy than the other states. This suggests that, even though opponent-state information is added for fine-tuning, the policy fails to fully exploit this information and instead relies predominantly on the task-dependent states. As explained in Section 5.1, in the tennis task — where adjusting the hitting motion according to the opponent's state is crucial — both the task state and the opponent state play important roles. Therefore, as indicated by the results in Fig. 3, it is likely that method (iii) exhibits unstable learning in the tennis task.

