# OpenReview forum: "LMA: Latent Motion Adjuster for Physics-based Multi-agent Interaction"
_TMLR — Accepted by TMLR_

### Review · Reviewer_aNTD · 2026-03-17

**Summary Of Contributions:**

## Summary

The paper proposes Latent Motion Adjuster (LMA), a small opponent-conditioned module that adds a residual correction to the latent action produced by a pretrained single-agent high-level policy before decoding through a motion prior. The intended benefit is to reuse single-agent skills in multi-agent interaction, rather than relearning interactive behavior from scratch. Experiments are conducted on SMPLOlympics table tennis and tennis, in both cooperative and competitive settings, using a PULSE-style latent motion prior and high-level policy.

## Strengths

- The method is simple, targeted, and easy to understand.

- Reusing a pretrained latent controller for multi-agent adaptation is a sensible and practically relevant goal.

- The cooperative-task results suggest a real early-learning benefit over some baselines.

- The paper includes mechanism-oriented analysis beyond reward curves, especially the sensitivity analysis and return-location visualizations.

## Weaknesses

- The submission is very close to prior work on hierarchical skill reuse in physics-based control and to residual-policy ideas, now applied as an additive correction in latent space rather than action space. Related discussion should more directly position the method against PULSE [1], multi-agent/team-play control [2], two-player sports control [3], and residual policy/control methods [4, 5].

- The empirical scope only includes two closely related racket-sport environments from the same benchmark and the same overall control stack. That is not enough to support the paper’s broader framing as a general mechanism for transferring single-agent skills to multi-agent RL.

- Several comparisons are not cleanly matched. In particular, some methods use random ball targets while others use a strategic network, which makes it hard to attribute gains to LMA itself rather than to differences in task-level decision support.

- The strongest missing baseline is an opponent-conditioned fine-tuning baseline that simply expands the high-level policy input with opponent state. The current “fine tuning” baseline appears artificially weak because it does not use the extra interaction observation.

## References

[1] Universal Humanoid Motion Representations for Physics-Based Control. Zhengyi Luo et al. 2024.

[2] From Motor Control to Team Play in Simulated Humanoid Football. Siqi Liu et al. 2022.

[3] Control Strategies for Physically Simulated Characters Performing Two-Player Competitive Sports. Jungdam Won et al. 2021.

[4] Residual Policy Learning. Tom Silver et al. 2019.

[5] Residual Reinforcement Learning for Robot Control. Tobias Johannink et al. 2019.

[6] Lifelike Agility and Play in Quadrupedal Robots Using Reinforcement Learning and Generative Pre-trained Models. Lei Han et al. 2024.

**Audience:**

Yes

**Audience Explanation:**

Researchers working on reusable humanoid skills, physics-based animation/control, and multi-agent adaptation would likely find the idea worth knowing, especially given the recent interest in universal motion representations, sports benchmarks for simulated humanoids, and transferring low-level skills into team-play or interactive settings.

**Claims And Evidence:**

No

**Claims Explanation:**

I can tell the paper shows a promising task-specific effect: a lightweight opponent-conditioned residual in latent space can help adapt a pretrained single-agent controller in these two sports settings.

However, I do not think the broader claims are yet established. The paper argues for a principled mechanism for transferring single-agent skills to multi-agent interaction, but the evidence is limited to a narrow benchmark family built for sports humanoids, with one motion-prior backbone and only three training trials per method. SMPLOlympics is a strong benchmark, but success there alone is not enough to justify the paper’s broader language. The baseline design weakens causal interpretation. The comparison mixes changes in latent adaptation, strategic target generation, and training/freezing choices. Because [6] already argued for a hierarchical reuse pipeline with a higher-level strategic module, the paper needs much cleaner isolation of what LMA adds beyond that line of work.

Meanwhile, the paper also underplays relevant conceptual overlap with residual methods, where a learned policy adds a correction on top of an existing controller. Since LMA is exactly an additive correction, just in latent space, this comparison is central rather than peripheral.

**Requested Changes:**

- Critical: Add a fair opponent-conditioned fine-tuning baseline where the pretrained high-level policy is expanded to consume opponent state and is then fine-tuned. This is the most direct alternative to LMA.

- Critical: Add a matched action-space residual baseline and, ideally, a residual-at-hidden-layer baseline. The paper currently does not demonstrate that the benefit is specific to latent-space modulation.

- Critical: Make all comparisons apples-to-apples with respect to ball target generation. Either all methods should use random targets or all should use the same strategic module.

- Critical: Clarify the optimization procedure for each variant. The pseudo-code should explicitly state which parameters are updated in the freeze and no-freeze settings.

- Strengthening: Report parameter counts, wall-clock training cost, and inference cost for all methods in the main paper.

---

> ### Author Response · Authors · 2026-04-13
> **Author Response (Part I)**
>
> Thank you for your constructive and insightful feedback. We appreciate the specific suggestions you provided for clarifying the positioning of our work relative to prior research. Below, we present our responses to the points you raised. In addition, the sections revised in response to the reviewers’ comments are shown in red in the manuscript. We believe that the revisions have strengthened the distinctions between prior work and our method, as well as the structured experimental evaluations demonstrating the effectiveness of our approach.
> - Requested Change 1:
> > Because [6] already argued for a hierarchical reuse pipeline with a higher-level strategic module, the paper needs much cleaner isolation of what LMA adds beyond that line of work.
> > Meanwhile, the paper also underplays relevant conceptual overlap with residual methods, where a learned policy adds a correction on top of an existing controller. Since LMA is exactly an additive correction, just in latent space, this comparison is central rather than peripheral.
> - Action Taken 1:
>   - To clarify the novelty of the proposed method, we have explicitly described the essential differences from residual learning in Section 2.2, as well as the distinctions from [6] in the second paragraph of Section 2.1. We show that residual adaptation in the latent motion space enables efficient skill reuse in multi-agent interactions without compromising stability or representational capacity.
> - Requested Change 2:
> > Critical: Add a fair opponent-conditioned fine-tuning baseline where the pretrained high-level policy is expanded to consume opponent state and is then fine-tuned. This is the most direct alternative to LMA.
> - Action Taken 2:
>   - We agree with your comment and conducted additional experiments to enable a fair comparison as latent adaptation based on opponent states. Specifically, we added a fair opponent‑conditioning fine‑tuning baseline as Baseline Method (iii) in Section 4.3, and we incorporated the results into Chapter 5 and Appendix A.10.  In baseline (iii), because all input information is fed into the high-level policy simultaneously, the additional opponent-state information tends to be underestimated. In contrast, the proposed method enables a clear division of roles, where the LMA responds to opponent states and the high-level policy responds to task states. Consequently, it achieves stable learning based on opponent-state information.
> - Requested Change 3:
> > Critical: Add a matched action-space residual baseline and, ideally, a residual-at-hidden-layer baseline. The paper currently does not demonstrate that the benefit is specific to latent-space modulation.
> - Action Taken 3:
>   - To clarify the effectiveness of residual reinforcement learning in the latent space, we incorporated the suggested baselines that apply residual learning in the action space and in the hidden layers as Baseline Methods (iv) and (v), respectively. We also reported the corresponding experimental results throughout Section 5. Furthermore, as noted in our first response, we explicitly cited prior work in the related‑work section (Section 2.2) and clearly described the differences between our proposed method and existing studies.
> - Requested Change 4:
> > Critical: Make all comparisons apples-to-apples with respect to ball target generation. Either all methods should use random targets or all should use the same strategic module.
> - Action Taken 4:
>   - To ensure a fair comparison, all experiments reported in the main text were conducted under the random‑target condition. In addition, the evaluation results for the strategy‑network condition were separated into Appendix A.7.
> - Requested Change 5:
> > Critical: Clarify the optimization procedure for each variant. The pseudo-code should explicitly state which parameters are updated in the freeze and no-freeze settings.
> - Action Taken 5:
>   - We have incorporated your comment into Algorithm 3. In addition, we have clarified in the final paragraph of Section 3.3 that, in the multi‑agent interaction stage of the proposed method, both configurations—with and without freezing the high‑level policy—are supported.
> - Requested Change 6:
> > Strengthening: Report parameter counts, wall-clock training cost, and inference cost for all methods in the main paper.
> - Action Taken 6:
>   - We have added the details of the hyperparameters used for each method to Appendix A.3. In addition, the inference‑time computational cost is reported in Appendix A.4.

---

> > ### Author Response · Authors · 2026-04-13
> > **Author Response (Part II)**
> >
> > - Requested Change 7:
> > > the evidence is limited to a narrow benchmark family built for sports humanoids, with one motion-prior backbone and only three training trials per method.
> > - Action Taken 7:
> >   - We agree with the reviewer’s observation that our evaluation is limited to ball-related tasks. However, we believe the proposed method is not restricted to such tasks and can be broadly applied. The motion prior used in our approach is trained on diverse human motion datasets, making it applicable not only to racket movements shown in this paper but also to a wide range of downstream tasks. Furthermore, the inputs to both the task policy and LMA consist of ego-centric representations of positions and velocities for each body part of the agent, the manipulated object, and the opponent. We consider this input representation to be generic and suitable for various tasks. As a future direction, we are also considering applications to non‑ball tasks such as collaborative robot manipulation, and we have stated this in Chapter 6 as part of our future work.

---

### Review · Reviewer_LGNh · 2026-03-18

**Summary Of Contributions:**

The paper introduces a three-stage process for multi-agent learning, where
they first learn an encoder-decoder model to infer a latent space,
on which subsequently first a task policy to guide a single agent, and
subsequently (or jointly) an adjustment policy for multi-agent interactions
is trained.
The method shows considerable performance improvements over a set of baselines
in two environments from SMPLOlympics.
Key strengths are the well-motivated story, clear paper structure, and significant
performance improvements across a broad ablation of baselines.
Key weaknesses are that the novelty is primarily combinatorial (each piece exists
in the literature, including residual latent-space adaptation), the experimental
evaluation has several gaps (undertrained baselines, a missing $s^\text{mi}_t$-concatenation
baseline, limited environment diversity, unablated reward imbalance), and the
competitive results in tennis are weak.

**Audience:**

Yes

**Audience Explanation:**

The question of how to reuse pretrained single-agent skills for multi-agent
interactions in physics-based control is practically relevant to researchers
in character animation, robotics, and multi-agent RL. The experimental setting
(whole-body humanoid control in cooperative and competitive sports) is appealing,
and the Sobol sensitivity analysis is a nice contribution. The findings would
be of interest even though the evidence needs strengthening.

**Claims And Evidence:**

No

**Claims Explanation:**

Each individual piece of the pipeline is known in the literature. While motion
priors are discussed, the concept of residual RL is never explicitly acknowledged
(Johannink et al., 2019). Specifically, residual adaptation in latent spaces was
introduced by Rana et al. (2023), who are not acknowledged. Others, e.g.,
Xie et al. (2020), also learned latent representations of strategies. The
paper's contribution is thus primarily in the combination of prior work.

For TMLR, I would consider this limited theoretical contribution justifiable if the empirical evidence were
stronger; however, there are several limitations in the experimental evaluation:
- The baselines get barely any training time; as such, it is not surprising that
  methods training from scratch don't perform. Does the performance gap still
  exist asymptotically?
- In stage 3, $\pi_h$ is either frozen or learned jointly with π_LMA; a baseline
  that learns a policy simply by concatenating $s^\text{mi}_t$ is missing to show that
  the relevant performance gains come from the LMA vs. merely from having access
  to $s^\text{mi}$ information.
- The evaluation is limited to tennis and table tennis, i.e., two relatively
  similar environments.
- The reward is very imbalanced with $r^g_t\approx 1$ but $r_\text{match} \in \{-100,100\}$.
  How would the methods perform if this discrepancy were smaller?

___
Rana et al. (CoRL, 2023): _Residual Skill Policies: Learning an Adaptable Skill-based Action Space for Reinforcement Learning for Robotics_
Xie et al. (CoRL, 2021): _Learning Latent Representations to Influence Multi-Agent Interaction_

**Requested Changes:**

- Extend the theoretical discussion of the novelty wrt the provided references and adjust the novelty accordingly.
- Extend the empirical evaluation wrt the comments given above

---

> ### Author Response · Authors · 2026-04-13
> **Author Response**
>
> We appreciate the time and effort you have dedicated to providing insightful feedback on ways to strengthen our paper. Below we summarize the changes that we made based on your comments. In addition, the sections revised in response to the reviewers’ comments are shown in red in the manuscript. We believe that the revisions have clarified the differences between prior studies and our method, and have further strengthened the systematic experimental evaluations demonstrating the effectiveness of our approach.
> - Requested Change 1:
> > Extend the theoretical discussion of the novelty wrt the provided references and adjust the novelty accordingly.
> > - The baselines get barely any training time; as such, it is not surprising that methods training from scratch don't perform. Does the performance gap still exist asymptotically?
> - Action Taken 1:
>   - As you pointed out, training from scratch is time‑consuming and may be insufficient as a baseline for evaluating the effectiveness of the proposed method. To clarify the novelty of our approach, we explicitly identify residual reinforcement learning as the most directly related method in Section 2.2, and we confirmed that the novelty of our method—residual adaptation in the latent motion space—is clearly articulated throughout the manuscript. In contrast, our method performs opponent‑state‑specific residual adaptation on latent motion representations acquired in a single‑agent setting, enabling efficient skill reuse in multi‑agent interactions without degrading stability or representational capacity. We believe this constitutes a clear distinction from both residual reinforcement learning and prior work on motion priors discussed in Section 2.1. Furthermore, the method proposed by Xie et al. (2020) focuses on learning the dynamics of opponent strategies, which differs from our formulation that treats latent representations as the agent’s own skills. Inferring opponent strategies is an important direction for future work, and we have explicitly stated this point in the final paragraph of Section 5.2.
> - Requested Change 2:
> > Extend the empirical evaluation wrt the comments given above
> > - In stage 3,  is either frozen or learned jointly with π_LMA; a baseline that learns a policy simply by concatenating is missing to show that the relevant performance gains come from the LMA vs. merely from having access to information.
> - Action Taken 2:
>   - To enable a fairer comparison, instead of training entirely from scratch, we expanded the baselines by incorporating residual reinforcement learning that reuses previously acquired skills  (method (iv) and (v) in Section 4.3). This allows us to more clearly demonstrate the effectiveness of the proposed method. The corresponding results are reported in Chapter 5. As you also pointed out, we adopted the baseline with simple concatenation as a direct comparison. We conducted additional experiments using this baseline as Method (iii) in Section 4.3 and added the corresponding results in Chapter 5 and Appendix A.10.
> - Requested Change 3:
> > The evaluation is limited to tennis and table tennis, i.e., two relatively similar environments.
> - Action Taken 3:
>   - We agree with the reviewer’s observation that our evaluation is limited to ball-related tasks. As a future direction, we are also considering applications to non‑ball tasks such as collaborative robot manipulation, and we have stated this in Chapter 6 as part of our future work. At the same time, we believe that the skills required for learning table tennis and tennis differ substantially. The former involves fast rallies but requires almost no lower-body movement, whereas the latter demands whole-body control involving continuous locomotion. This difference changes the relative importance of the hitting agent and the receiving agent, and—as shown in the third paragraph of Section 5.1—significantly affects the results of both Baseline Method (iii) and our proposed method.
> - Requested Change 4:
> > The reward is very imbalanced with $r_t^g \approx 1$ but $r_{match} \in -100, 100$. How would the methods perform if this discrepancy were smaller?
> - Action Taken 4:
>   - We agree with the reviewer’s observation that the two rewards used in the adversarial task, $r_t^{g}$ and $r_t^{match}$, differ significantly in scale. However, we believe that, when viewed over the course of an entire episode, the balance between these rewards is reasonably maintained. Since the former is received at every timestep, whereas the latter is given only at the moment when the match outcome is determined, the magnitude of the latter must be larger in order to achieve an appropriate balance. Moreover, the total amount of $r_t^g$ accumulated during a rally typically falls within the range of 50 to 150, which is not substantially different from the value of $r_t^{match}$.

---

### Review · Reviewer_75qv · 2026-03-30

**Summary Of Contributions:**

## Summary
The work proposes LMA (Latent Motion Adjuster), a framework designed to bridge the gap between single-agent skill acquisition and complex multi-agent interactions in physics-based simulations. Rather than learning multi-agent behaviors from scratch—which is often inefficient—the authors suggest reusing pretrained single-agent policies and adapting them to the presence of other agents through structured modulation in a latent motion space. It formulates multi-agent interaction as a latent adaptation problem, allowing for the direct reuse of pretrained single-agent skills, and establishes a clear progression from pretraining a motion prior (Stage 1), to learning a single-agent task policy (Stage 2), and finally adapting that policy for multi-agent interaction using LMA (Stage 3). Demonstrates through physics-based benchmarks that this approach significantly improves sample efficiency and interaction performance compared to fine-tuning or learning from scratch.

## Strengths
1. Efficiency and Reuse: By leveraging pretrained skills, the model avoids the "relearn from scratch" bottleneck common in MARL, leading to much faster convergence in cooperative and competitive tasks.
2. Generalizability: The framework uses generic input representations and is shown to be effective across qualitatively different tasks, suggesting it can be applied to a wide range of embodied interaction tasks.
3. Interpretability: Through Sobol sensitivity analysis, the authors demonstrate that the LMA selectively focuses on relevant opponent-state dimensions (like racket height) to adjust behavior.
4. Low Computational Overhead: Both the task policy and LMA are lightweight MLPs, adding negligible computation time compared to the physics simulation itself.

## Weaknesses
1. Lack of Strategic Inference: The current model reacts to the opponent's immediate state but does not infer long-term strategies, making it difficult to handle anticipatory actions like moving forward for a predicted drop shot.
2. Dependence on Single-Agent Quality: Performance in the multi-agent stage is heavily constrained by the diversity and quality of the skills learned in the single-agent stage; if a motion was never learned (e.g., returning a ball from the endline), the agent cannot adapt to it later.
3. Hyperparameter Sensitivity: The effectiveness of the LMA is highly dependent on the adjustment scale ($\eta$), and the optimal value changes depending on whether the high-level policy is frozen or trained jointly.

**Audience:**

Yes

**Audience Explanation:**

The findings of this paper address core challenges in reinforcement learning and embodied AI that are highly relevant to the TMLR audience, particularly those specializing in robotics, multi-agent systems, and representation learning.

**Broader Impact Concerns:**

No concerns about the ethical implications of the work.

**Claims And Evidence:**

Yes

**Claims Explanation:**

The submission provides accurate and convincing evidence to support its claims through a combination of quantitative performance metrics, statistical validation, and qualitative behavioral analysis across diverse physics-based environments.

**Requested Changes:**

1. Clarification of Strategy Inference vs. Immediate State: The authors state that the model does not infer opponent strategy, which leads to suboptimal global behaviors like failing to anticipate drop shots. A critical addition would be a brief discussion or a small-scale experiment showing if the LMA can be conditioned on a short history of opponent states (e.g., using a GRU or Transformer) rather than just the immediate state $s_t^{mi}$. This would clarify if the limitation is architectural or fundamental to the latent adaptation approach.
2. Formal Definition of $\eta$ Scheduling: The paper mentions that $\eta$ was set to 0.0001 and that another value was annealed from 0.01 to 0.001. There is a typographical omission in the text regarding which parameter was annealed. Clarifying this schedule is vital for reproducibility, especially given the sensitivity analysis in Section A.4.
3. Discussion on Skill Diversity: The results indicate that if a skill (like returning from the endline) wasn't in the single-agent phase, the agent cannot perform it in the multi-agent phase. A stronger version of this paper would discuss how the "coverage" of the initial latent space $z$ limits the "expressivity" of the LMA.
4. Ablation of the LMA Architecture: The LMA is currently a two-layer MLP. A brief mention of whether a simpler linear adjustment or a deeper network was tested would provide better architectural guidance for future work.
5. Visualizing the Latent Shift: It would be beneficial to provide a 2D visualization (e.g., t-SNE or PCA) of the latent space showing the "base" latent $z_t$ from the task policy and the "adjusted" latent $z'_t$ after LMA modulation. This would visually confirm that the adjustments are indeed "fine-grained" as claimed.

---

> ### Author Response · Authors · 2026-04-13
> **Author Response**
>
> Thank you for providing these insights. Below, we present our responses to the reviewer’s comments. In addition, the sections revised in response to the reviewers’ comments are shown in red in the manuscript.
> - Requested Change 1:
> > Clarification of Strategy Inference vs. Immediate State: The authors state that the model does not infer opponent strategy, which leads to suboptimal global behaviors like failing to anticipate drop shots. A critical addition would be a brief discussion or a small-scale experiment showing if the LMA can be conditioned on a short history of opponent states (e.g., using a GRU or Transformer) rather than just the immediate state $s_t^{mi}$. This would clarify if the limitation is architectural or fundamental to the latent adaptation approach.
> - Action Taken 1:
>   - We agree with the comment that the opponent’s state history can contribute to strategy inference. To examine how the length of the opponent‑state observation affects the proposed method, we conducted additional experiments and reported the results in Appendix A.8.
> - Requested Change 2:
> > Formal Definition of $\eta$ Scheduling: The paper mentions that $\eta$ was set to 0.0001 and that another value was annealed from 0.01 to 0.001. There is a typographical omission in the text regarding which parameter was annealed. Clarifying this schedule is vital for reproducibility, especially given the sensitivity analysis in Section A.4.
> - Action Taken 2:
>   - We have added details of the annealing procedure in Appendix A.3. As the previous description was unclear, we apologize for the confusion. The annealing is applied to $\beta$ during pre‑training, and it is not applied to $\eta$, which is used when training LMA. Accordingly, the description of $\beta$‑annealing has been moved from Section 4.2 to Appendix A.3. Although $\eta$ is not annealed, we describe in Appendix A.5 how different values of $\eta$ influence the learning process.
> - Requested Change 3:
> > Discussion on Skill Diversity: The results indicate that if a skill (like returning from the endline) wasn't in the single-agent phase, the agent cannot perform it in the multi-agent phase. A stronger version of this paper would discuss how the "coverage" of the initial latent space $z$ limits the "expressivity" of the LMA.
> - Action Taken 3:
>   - We have added results demonstrating the coverage of the latent motion space in Appendix A.9. Specifically, Fig. 19 demonstrates that the initial latent motions differ, and Fig. 17 shows that the learning of multi-agent interactions is strongly influenced by these initial latent motions.
> - Requested Change 4:
> > Ablation of the LMA Architecture: The LMA is currently a two-layer MLP. A brief mention of whether a simpler linear adjustment or a deeper network was tested would provide better architectural guidance for future work.
> - Action Taken 4:
>   - We have presented the effect of the number of LMA network layers on learning performance in Fig. 14 of Appendix A.6.
> - Requested Change 5:
> > Visualizing the Latent Shift: It would be beneficial to provide a 2D visualization (e.g., t-SNE or PCA) of the latent space showing the "base" latent $z_t$ from the task policy and the "adjusted" latent $z'_t$ after LMA modulation. This would visually confirm that the adjustments are indeed "fine-grained" as claimed.
> - Action Taken 5:
>   - We have incorporated results demonstrating the limited adjustment range of LMA into Fig. 18 in Appendix A.9.

---

### Review · Reviewer_DdrU · 2026-04-01

**Summary Of Contributions:**

The authors develop a new method to use learned complex latent skills in multi-agent environments, adapted to physics-based motion interactions. Towards this goal, a three-stage training pipeline is proposed, where agents learn latent skills individually, and a Latent Motion Adjuster (LMA) adapts those skills to the multi-agent setting. Compared to previous work, this pipeline doesn’t require a heavy multi-agent training from scratch, and allows for reuse of skills learned by single agent policies.

Strengths:

- Clear motivation and presentation of the proposed method
- The method is simple and achieves strong results in the experiments
- The approach is sensible for practical applications

Weaknesses:

- Table and table tennis tasks are very similar. It would be interesting to know how the method generalized to other environments beyond those two.

I must mention that I am not familiar enough with motion control to measure the novelty of the paper, but other reviewers have flagged that some relevant previous work might be missing in the paper.

**Audience:**

Yes

**Audience Explanation:**

Learning latent skills for complex motions is of great importance, in the single and multi-agent settings. The curse of dimension of multi-agent systems motivates the need for lightweight adaptations of policies learn from single-agent training, and the paper presents such adaptation through the concept of Latent Motion Adjuster (LMA).

**Broader Impact Concerns:**

Nothing to be mentioned in particular.

**Claims And Evidence:**

Yes

**Claims Explanation:**

The overall method, is clear and well motivated, and shows improved performance in practice. I should note, however, that the pipeline is shown to outperform several baselines only across two similar environments, which limits the scope of the contribution.

**Requested Changes:**

I find the paper quite clear in general. My comments would be on expanding on the novelty of the work as well as expanding on these two comments:

- “the rotation representation uses (Zhou et al., 2019).” Can you briefly summarize this rotation representation in the paper? This is to give some pointers/understanding about the method without having to look at Zhou et al. (2019).
- “The study by (Han et al., 2024) is similar […] our method directly adjusts the latent motions generated by single-agent policies.” Please clarify the weaknesses of the method from Han et al (2024).

Regarding the experiments, I agree that the two environments are a bit limited. How hard would it be to introduce a more diverse set of experiments? Or, to which level are the two environments different in the task they solve?

Minor:

- Figures 5 and 6 are a little hard to read.

---

> ### Author Response · Authors · 2026-04-13
> **Author Response**
>
> We appreciate the valuable feedback commented by the reviewer. We will answer your comments below. In addition, the sections revised in response to the reviewers’ comments are shown in red in the manuscript.
> - Requested Change 1:
> > “the rotation representation uses (Zhou et al., 2019).” Can you briefly summarize this rotation representation in the paper? This is to give some pointers/understanding about the method without having to look at Zhou et al. (2019).
> - Action Taken 1:
>   - We added the following sentence to the paragraph in Section 4.1. Since the procedure is complex, we describe only its high‑level idea:
>   > “That is, we convert rotation matrices into a compact and continuous 6D representation through an orthogonalization process.”
> - Requested Change 2:
> > “The study by (Han et al., 2024) is similar […] our method directly adjusts the latent motions generated by single-agent policies.” Please clarify the weaknesses of the method from Han et al (2024).
> - Action Taken 2:
>   - We have clarified the limitations of the method proposed by Han et al. (2024) and the differences addressed by our proposed approach at the end of Section 2.1.
> - Requested Change 3:
> > Regarding the experiments, I agree that the two environments are a bit limited. How hard would it be to introduce a more diverse set of experiments? Or, to which level are the two environments different in the task they solve?
> - Action Taken 3:
>   - We agree with the reviewer’s observation that our evaluation is limited to ball-related tasks. As a future direction, we are also considering applications to non‑ball tasks such as collaborative robot manipulation, and we have stated this in Chapter 6 as part of our future work. At the same time,  we believe that the skills required for learning table tennis and tennis differ substantially. The former involves fast rallies but  requires almost no lower-body movement, whereas the latter demands whole-body control involving continuous locomotion. This difference changes the relative importance of the hitting agent and the receiving agent, and—as shown in the third paragraph of Section 5.1—significantly affects the results of both the additional fine‑tuning baseline with expanded input dimensions (Method (iii)) and our proposed method.
> - Requested Change 4:
> > Figures 5 and 6 are a little hard to read.
> - Action Taken 4:
>   - We improved the readability of Figures 5 and 6 by enlarging their sizes and increasing the spacing between lines.

---

### Decision · Action_Editor_cxJm · 2026-05-12

**Recommendation:** Accept with minor revision

**Additional Comments:**

N/A

**Audience:**

Yes

**Audience Explanation:**

Skill adaptation in multi agent systems is a topic of large interests in the community.

**Claims And Evidence:**

Yes

**Claims Explanation:**

The authors design a new algorithm to perform skills in multi-agent environments using a shared latent space. All reviewers agree that the idea, while conceptually simple, is sound. Furthermore, through the author's rebuttal, extensive updates to the manuscript have aided its clarity. In particular, the rebuttal successfully addressed the chief concern that the latent-space adaptation was the principal driver of the results. This represents clearer evidence to the method.